# Rapid generation of hypomorphic mutations

Laura L. Arthur[1], Joyce J. Chung[2], Preetam Jankirama[3,4], Kathryn M. Keefer[1], Igor Kolotilin[5], Slavica Pavlovic-Djuranovic[1], Douglas L. Chalker[2], Vojislava Grbic[3], Rachel Green[6], Rima Menassa[4], Heather L. True[1,7], James B. Skeath[8] & Sergej Djuranovic[1]

Hypomorphic mutations are a valuable tool for both genetic analysis of gene function and for synthetic biology applications. However, current methods to generate hypomorphic mutations are limited to a specific organism, change gene expression unpredictably, or depend on changes in spatial-temporal expression of the targeted gene. Here we present a simple and predictable method to generate hypomorphic mutations in model organisms by targeting translation elongation. Adding consecutive adenosine nucleotides, so-called polyA tracks, to the gene coding sequence of interest will decrease translation elongation efficiency, and in all tested cell cultures and model organisms, this decreases mRNA stability and protein expression. We show that protein expression is adjustable independent of promoter strength and can be further modulated by changing sequence features of the polyA tracks. These characteristics make this method highly predictable and tractable for generation of programmable allelic series with a range of expression levels.

[1] Department of Cell Biology and Physiology, Washington University School of Medicine, St Louis, Missouri 63110, USA. [2] Department of Biology, Washington University, St Louis, Missouri 63105, USA. [3] Department of Biology, The University of Western Ontario, 1151 Richmond Street, London, Ontario, Canada N6A5B7. [4] Science and Technology Branch, Agriculture and Agri-Food Canada, 1391 Sandford Street, London, Ontario, Canada N5V4T3. [5] Scattered Gold Biotechnology Inc. 14 Denali Terrace, London, Ontario, Canada N5X 3W2. [6] Department of Molecular Biology and Genetics, Howard Hughes Medical Institute, Johns Hopkins University School of Medicine, 725 North Wolfe Street, Baltimore, Maryland 21205, USA. [7] The Hope Center for Neurological Diseases, Washington University School of Medicine, St Louis, Missouri 63110, USA. [8] Department of Genetics, Washington University School of Medicine, St Louis, Missouri 63110, USA. Correspondence and requests for materials should be addressed to S.D. (email: sergej.djuranovic@wustl.edu).

Manipulation of gene activity is a standard genetic approach to gain insight into gene function in single and multicellular organisms[1]. In many cases, complete loss of gene function (null allele or knockout of the locus) will provide the most valuable information about gene function. However, for essential genes, complete loss of function leads to lethality, which usually precludes obtaining functional information for later cellular or developmental stages. Similarly, for genes that function in multiple cellular/developmental processes and have pleiotropic null mutant phenotypes, it can be difficult to distinguish primary from secondary effects. In many of these cases, however, partial loss of function or hypomorphic mutations can overcome lethality and pleiotropy, allowing later stage cells and organisms to be examined for phenotypic consequences. Furthermore, hypomorphic mutations, because they retain residual gene activity and partial phenotypes, are used in suppressor or enhancer genetic screens to identify other genes that act in the same biological process. One example of a hypomorphic condition is a 50% reduction in gene activity from heterozygosity for a null allele, which for some genes can display mutant phenotypes (called a haploinsufficency). However, for other genes a 50% reduction in gene activity is sufficient for normal function, and thus hypomorphic mutations with a further reduction are required to observe a phenotype. For these reasons, it is important to have the ability to generate hypomorphic mutations with a range of loss of gene activity.

Hypomorphic mutations are traditionally obtained in forward genetic chemical mutagenesis screens. The hypomorphic allele needs to be isolated, identified, and then genetically and biochemically characterized in order to be further used in analysis to deduce gene function. As a consequence, this process is time consuming. In addition, because of evolution, discovered alleles can be species specific. These difficulties and the importance of hypomorphic alleles have prompted the development of several methods to generate hypomorphic mutations directly in various model organisms[2–11]. However, these methods again are usually specific to one organism, can have unpredictable alterations in gene activity (separation of function, gain of function), or can change other aspects of regulation that affect interpretation of phenotype, such as spatial-temporal gene expression. With the rise of gene editing systems, such as CRISPR/Cas9 (ref. 12) or TALEN technology[13], and more extensive use of synthetic and systems biology approaches[14,15], there is an increasing interest in generating hypomorphic mutations of a target gene through simpler, more systematic and rapid approaches.

Here we present a method for the generation of hypomorphic mutations that produces a range of reduced levels of nearly wild type protein through the use of disrupted translational elongation. This method is based on polyA tracks, a novel cis regulatory element that decreases gene expression by disrupting messenger RNA (mRNA) translation[16,17]. Insertion of consecutive adenosine nucleotides into the open reading frame of an mRNA will decrease protein expression by decreasing the efficiency of the translation elongation phase leading to diminished production of protein and mRNA destabilization, and thus to diminished mRNA levels.

## Results

**Use of polyA tracks to generate hypomorphic mutants.** We have recently identified polyA tracks as a regulator of gene expression[16,17]. This mechanism is used endogenously in most eukaryotic genomes and regulates ~2% of human genes[16,18]. The polyA track causes ribosomal stalling and frameshifting during translation elongation, leading to mRNA instability and degradation of nascent protein products[16,17]. The translation elongation cycle is an ideal target for a universal method of gene regulation because it is the most highly conserved step in protein biosynthesis between bacteria and eukaryotes[19]. Therefore, we reasoned that polyA tracks, because of their versatility in lengths and sequence composition, can be used as a system to create programmable hypomorphic mutants and regulate gene expression in a wide variety of model organisms (Fig. 1).

We have generated a fluorescent reporter gene that has an insertion of defined polyA tracks in order to control the amount of expression (Fig. 1a). The reporter consists of either a constitutive or inducible promoter driving expression of the mCherry fluorescence and other reporter proteins. A double HA-tag was added at the beginning of the coding sequence for detection through western blot analysis. The polyA track is inserted directly after the HA-tag. The length of the polyA track varies from 9 to 36 consecutive adenosine nucleotides, adding 3–12 lysine residues to the protein sequence (Fig. 1a). To assess the effects of polybasic peptide arising from sequential lysine residues[20,21], we also generated control reporters with consecutive lysine AAG codons. We hypothesized that as the length of the polyA track is increased, expression of the reporter gene products will decrease (Fig. 1b,c). These reporters can be transiently transfected, recombined or inserted into the genome of cell cultures or whole organisms. Likewise, endogenous genes can be edited to include a polyA track in their open reading frames (ORF's) using genome editing methodology.

**PolyA tracks can regulate gene expression in _Escherichia coli_.** We first tested whether polyA tracks can be used in single-cell model organisms to attenuate gene expression from a defined reporter gene. Previous attempts to control gene expression in _Escherichia coli_ cells have used degeneration of ribosome binding sequence[2] (rbs) or spacer sequence between rbs sequence and starting AUG codon[22], but did not focus on the coding sequence. To show that polyA tracks can be used to control gene expression in _E. coli_ cells, we created a set of reporters with increasing length of polyA tracks under the arabinose-inducible promoter pBAD (Supplementary Fig. 1). We transformed _E. coli_ cells with plasmids expressing HA-mCherry, HA-(AAG)$_n$-mCherry or HA-(AAA)$_n$-mCherry. All _E. coli_ cell cultures were induced at the same optical density and monitored for both cell growth and fluorescence of the mCherry constructs during induction. While _E. coli_ containing wild type and Lys$_{AAG}$ controls (6×, 9× and 12× lysine AAG codons) show no significant differences in the amount of mCherry fluorescence, cell cultures containing constructs with polyA tracks show progressively less fluorescence with increasing length of the polyA track (Fig. 2a). Addition of 9 and 12As in a row (3 or 4 Lys$_{AAA}$ codons) consistently reduced fluorescence of mCherry reporter by 15–35%. Further additions in the length of the polyA track resulted in continuing decreases in mCherry reporter fluorescence to the point where 36 consecutive adenosine nucleotides resulted in barely visible expression of the reporter (<5% of wild type). Western blot analyses of equal amounts of _E. coli_ cell lysates expressing different polyA track and control reporters were consistent with the mCherry fluorescence data and indicated again that protein abundance of reporter proteins strongly depends on the length of the polyA track (Fig. 2b). Reporters with 9 and 12As in a row (3 and 4Lys$_{AAA}$ codons, respectively) show reduction in protein abundance in the range of 20–40% of the wild type mCherry, and constructs with >27As in a row (9 and more Lys$_{AAA}$ codons) were nearly undetectable by western blot analyses.

To further fine tune expression of the reporter gene with polyA tracks, we tested whether insertion of non-lysine codons in the middle of long polyA tracks (33As, 11 AAA codons) would result

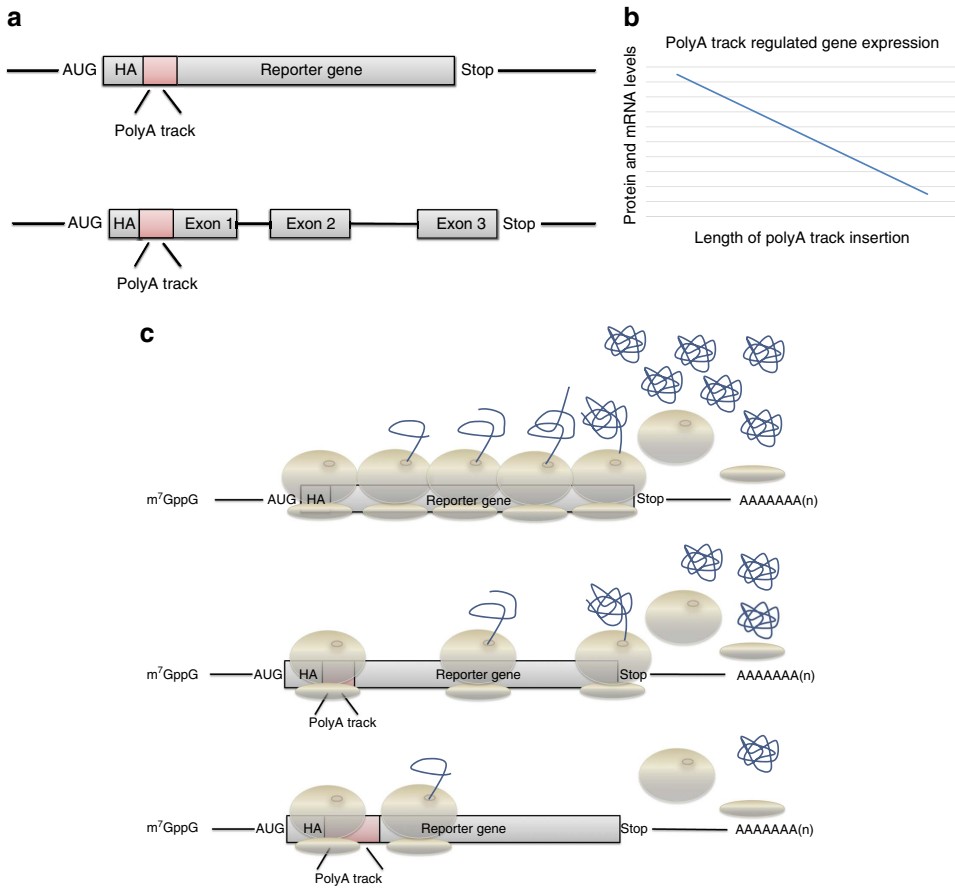

**Figure 1 | Design and mechanism of polyA track tag regulated gene expression.** (**a**) Scheme of inserted polyA tracks in the reporter genes used in this study. Hemagglutinin (HA) tag (grey) and polyA tracks (red) were introduced in the coding region of the reporter genes next to the start AUG codon. Exon boundaries as well as termination codon (Stop) are indicated. (**b**) Proposed correlation between gene products levels, mRNA and protein, and the length of inserted polyA track tags. The reduction in levels of both reporter protein and mRNA is dependent on increasing length of consecutive adenosine nucleotides in the coding sequence. (**c**) Scheme of translation of eukaryotic reporter mRNA with or without inserted polyA tracks. The length of inserted polyA track tag determines the protein output of the regulated reporter gene as indicated by the number of globular protein structures. Features of the eukaryotic mRNAs (m7GpppG—cap, AUG—start codon, Stop—termination codon and polyA tail), as well as HA-tag, position of the polyA track tag, ribosome and nascent polypeptide chain are illustrated in the scheme.

in change of the reporter accumulation in *E. coli* cells. Indeed, addition of such codons did not change levels of the fluorescent reporter drastically but fine-tuned them to the fluorescent values between constructs with 30As or 33As in a row (Supplementary Fig. 2). Additional analysis of XAA and AAY codons, where X and Y denote C/G or T/C/G nucleotides respectively, for programming the length of polyA tracks revealed that gene attenuation by polyA tracks does not depend necessarily on lysine codons but other codons can contribute to the overall length and effects of polyA tracks (Supplementary Fig. 2). In a similar manner endogenous genes with different arrangement of A–rich codons (mainly lysine encoding AAA and AAG codons) have shown differential gene regulation in our previous study[17].

**PolyA tracks can be used in protozoan *T. thermophila*.** We previously showed that polyA tracks can influence expression of the reporter genes in *Saccharomyces cerevisiae* cells[17]. To test whether polyA tracks can regulate gene expression in another model, single-cell eukaryotic organism, we monitored the effect of various length tracks on YFP expression in the protozoan *Tetrahymena thermophila*. The genome of *T. thermophila* has extremely high AT content ($>75\%$) and has been extensively used as a microbial animal model[23]. Our *T. thermophila* reporter

contained the coding sequence of a *Macronucleus-Localized Protein* of unknown function (*MLP1*, TTHERM_00384860) fused to eYFP protein (Supplementary Fig. 3). The fusion with MLP1 directed YFP to *Tetrahymena* macronuclei to allow easier quantification of YFP levels (Fig. 2c). These two proteins were fused, separated by linkers containing an HA-tag (MLP1-HA-YFP (WT)) and polyA tracks of 18, 27 or 36As, $(AAA)_6$, $(AAA)_9$ or $(AAA)_{12}$, respectively, or 12 $Lys_{AAG}$ $(AAG)_{12}$ codons inserted as a control. All constructs were expressed upon cadmium-induction of the upstream *MTT1* promoter. Just as in our *E. coli* experiments with mCherry reporter, the YFP gene containing increasing lengths of polyA tracks exhibited a progressive decrease in total protein accumulation, measured by fluorescence, relative to the HA-linker fusion or $Lys_{AAG}$ insertion controls (Fig. 2d). The construct with 18As in a row ($6Lys_{AAA}$) showed ∼50% reduction in protein fluorescence, while constructs with 27 and 36As exhibited nearly undetectable levels of fluorescence and required 20 times longer exposure for detection of YFP by microscopy. The construct with $12Lys_{AAG}$ codons showed fluorescence that was 4–5 fold lower than the WT construct. This effect was comparable to the polybasic peptide stalling that was observed earlier in *S. cerevisiae* cells[17,20,21]. We further supported our fluorescence results by western blot analyses (Fig. 2e). The MLP1-HA-YFP-fusion (WT)

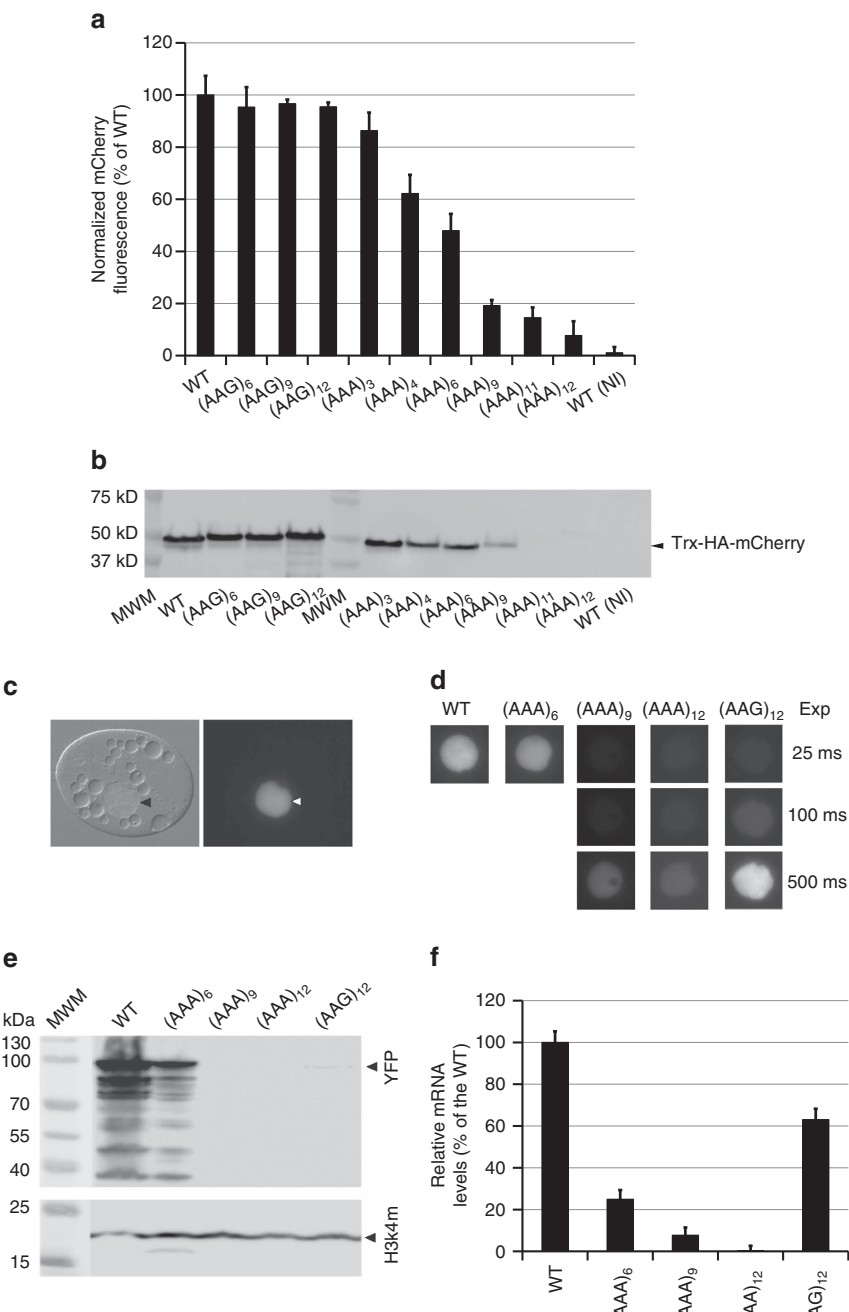

**Figure 2 | Regulation of reporter gene by polyA tracks in the single-cell prokaryotic and eukaryotic organisms.** (**a**) Percentage of mCherry fluorescence of tested Lys$_{AAG}$ ((AAG)$_{6-12}$) and Lys$_{AAA}$((AAA)$_{3-12}$) insertion constructs compared with wild type fluorescence (WT, no insertion construct). mCherry fluorescence was assayed at excitation wavelength of 475 ± 9 nm and emission was detected at 620 ± 9 nm. Error bars indicate mean mCherry fluorescence values ± s.d. for three individual *E. coli* colonies for each construct. Background levels of mCherry expression can be estimated from the fluorescence of the non-induced wild type construct (WT(NI)). (**b**) Western blot analysis of mCherry constructs expressed in *E. coli* cells. Equal amounts of *E. coli* cell lysates with Thioredoxin(Trx) fusion proteins were used for analysis. Fusion proteins were detected using HA-tag specific antibody. Positions of the fusion protein (Trx-HA-mCherry) and sizes of molecular weight markers (MWM) are indicated. (**c**) Representative differential interference contrast microscopy (left panel) and the corresponding fluorescence image (right panel − 25 ms exposure) of a *T. thermophila* cell expressing the wild type (WT) MLP1-HA-YFP fusion. Arrowheads denote the position of the macronucleus. (**d**) *MPL1*-HA-YFP accumulation within macronuclei of live *T. thermophila* cells expressing an allelic series of fusion proteins—WT, (AAA)$_{6-12}$, and (AAG)$_{12}$—was visualized by epifluorescence microscopy. Different exposures times are indicated on the right to demonstrate the relative accumulation of each variant. (**e**) Western blot analysis was performed with whole cell lysates made from *T. thermophila* cells expressing the MLP-HA-YFP fusion proteins. Protein from equivalent cell numbers was loaded in each lane and detected using YFP specific antibody (top panel) and normalized to the nuclear histone species, histone H3 trimethyl-lysine 4 (H3K4m) (bottom panel). Positions of the full-length fusion protein (YFP), normalization control (H3k4m), and sizes of molecular weight markers (MWM) are indicated. Degradation of excess fusion protein is readily apparent as faster migrating species below the full-length MLP1-HA-YFP. (**f**) Steady state levels of fusion gene constructs measured by qRT-PCR. Relative levels of the mRNA for (AAG)$_{12}$ and (AAA)$_{6-12}$ are presented as percentage of the wild type (WT) construct mRNA levels. Error bars represent mean ± s.d. values ($n = 3$).

was readily visible, whereas the polyA track-YFP fusions exhibited attenuated expression. Insertion of 18A's (6Lys$_{AAA}$) showed expression levels that were ~35% of the HA-YFP control, while 27 and 36A's (9 and 12Lys$_{AAA}$) constructs were at the limit of detection (Fig. 2e). Insertion of 12Lys$_{AAG}$ codons in the fusion protein resulted in a 6- to 8-fold reduction in expression of fusion protein. Since the different fusion proteins were all transcribed from the *MTT1* promoter under identical induction conditions we reasoned that the amount of mRNA produced for each construct should be equivalent. We used qRT-PCR to quantify the steady-state level of YFP mRNA for each fusion (Fig. 2f). The steady-state mRNA levels of polyA track-YFP constructs reflected the decreasing YFP protein accumulation relative to the WT consistent with earlier reports in *S. cerevisae*[17,20]. Insertion of 18As (6Lys$_{AAA}$ codons) reduced mRNA levels to approximately 30–35% of WT levels, while insertion of 27 and 36As (9 and 12Lys$_{AAA}$ codons, respectively) reduced mRNA levels to <5% of the HA-YFP construct (Fig. 2d). Interestingly, while attenuation at the protein level was stronger for the insertion of 12Lys$_{AAG}$ codons than for 6Lys$_{AAA}$ codons, the trend was not consistent at the mRNA level. The observed discrepancies between the protein levels and mRNA degradation in the case of polybasic peptide (12 AAG codon) are likely attributed to the two distinct pathways employed in regulation of polybasic peptide stalling and polyA track induced stalling and frameshifting in eukaryotic cells[16,17,24]. Nevertheless, the effects of polyA tracks on regulation of the reporter gene showed a consistent relationship between mRNA levels and protein accumulation levels. Most importantly, we were able to control expression of reporter genes using polyA tracks in *T. thermophila*, a single-cell AT-rich protozoan, as we previously reported in *S. cerevisiae* and the above-mentioned *E. coli* cells[17].

**PolyA tracks can regulate gene expression in plant tissues**. To test whether polyA tracks can attenuate gene expression in plants, we transiently co-expressed HA-(AAA)$_n$-mCherry with an YFP construct as an internal control in the model plant *Nicotiana benthamiana* (Supplementary Fig. 4). The expression of mCherry and YFP was assessed by fluorescence imaging (Fig. 3a). Like cell cultures, *N. benthamiana* epidermal cells showed attenuated mCherry fluorescence proportional to the length of the polyA tracks (6,9 and 12 Lys$_{AAA}$ codons) compared with the HA-mCherry and 12 Lys$_{AAG}$ control constructs (Fig. 3a). The fluorescence data for each construct revealed the same trend of gene expression regulation as in *T. thermophila* cells (Fig. 2d). As fluorescence in this assay was not quantifiable, protein abundance was determined by semi-quantitative western blot analysis of *N. benthamiana* leaves infiltrated with the HA-(AAA)$_n$-mCherry. The levels of HA-mCherry proteins were normalized to levels of the cis-linked selectable marker phosphinotricin acetyl transferase (BAR) in the same sample (Fig. 3b,c). The addition of a polyA track with 18As (6 Lys$_{AAA}$) decreased protein accumulation to ~70% of HA-mCherry levels. Further reduction of mCherry protein accumulation, to 30% and below the detection limit was observed in 9 Lys$_{AAA}$ and 12 Lys$_{AAA}$ constructs, respectively (Fig. 3c). Insertion of 12 Lys$_{AAG}$ codons displayed ~50% reduction in the reporter expression compared with WT construct. Parallel analyses of steady-state mRNA levels of transcripts with increasing lengths of polyA tracks showed progressively reduced levels of polyA track mRNAs when compared with transcript levels of the HA-mCherry and AAG-containing control constructs (Fig. 3d). mRNA levels were reduced to ~50–55% of WT expression for 6Lys$_{AAA}$ transcripts, while 9 and 12Lys$_{AAA}$ constructs had reduced mRNA levels to ~30 and 20% of control, respectively (Fig. 3d). Insertion of 12 Lys$_{AAG}$ codons exhibited

marginal effects on mRNA levels and again showed the already observed discrepancy between mRNA levels and protein accumulation as a result of polybasic peptide stalling. However, these results indicate once again that polyA tracks affect both mRNA and protein levels more consistently and can be used to regulate gene expression in plants.

**PolyA tracks can regulate genes in human tissue cultures**. To further assess the universality of polyA tracks on protein expression, we tested our reporter series in human tissue cultures using HeLa cells. Plasmids with HA-mCherry, HA-(AAG)$_{12}$-mCherry and HA-(AAA)$_n$-mCherry reporters, driven by the constitutively active CMV promoter, were electroporated into HeLa cells for transient expression. Protein abundances were assessed by western blot analyses 24 h after electroporation (Fig. 3e). As in our previous study on expression of endogenous and synthetic polyA tracks in various human tissue cultures[16], constructs with increasing length of polyA tracks (6, 9 and 12 Lys$_{AAA}$) were expressed at lower levels than control constructs and the reductions in protein expression were proportional to the length of polyA track. The construct with 18As (6Lys$_{AAA}$) displayed an ~3-fold reduction in expression compared with the WT construct. Insertion of 27 and 36As (9 and 12Lys$_{AAA}$, respectively), exhibited a 6- and 25-fold reduction of HA-mCherry expression compared with WT (Fig. 3f). The control construct with 12Lys$_{AAG}$ codons did not show any reduction in protein levels compared with the WT construct (Fig. 3e,f). These results again argue for differences between translational stalling induced by polybasic peptides[16,17,20,21], which seems to be cell- or organism-specific and unpredictable, and polyA track-induced ribosomal stalling and frameshifting[16,17], which is clearly dependent on the length of polyA tracks. Importantly, this latter pathway appears to be conserved between multiple organisms. Altogether with our previous study[16], our results indicate that polyA tracks can readily be used to regulate expression of reporters or genes transiently transfected in diverse eukaryotic tissues and cultured cell systems, such as *N. benthamiana* and human cell cultures, as well as other mammalian or insect tissue culture systems[16].

**PolyA tracks can regulate gene expression in model organisms**. We next sought to test whether polyA tracks can be used to regulate reporter gene expression in complex, multicellular organisms. We chose the fruit fly, *Drosophila melanogaster*, because of the well-developed tools in the manipulation of endogenous genetic loci, as well as for the ready assessment of the mCherry reporter activity. Using the PhiC31-integrase approach[25], we generated single transgene insertions of the HA-mCherry and HA-(AAG)$_{12}$-mCherry controls, and HA-(AAA)$_n$-mCherry (6, 9 and 12Lys$_{AAA}$) constructs in the identical genomic location in the third chromosome (see methods; Supplementary Fig. 5). All constructs contained an Upstream Activation Sequence (UAS) followed by the *HSP70* promoter which actively transcribes mCherry reporter mRNAs in response to expression of GAL4 protein[26]. To drive expression of mCherry in all tissues, each transgenic line was crossed to a driver line with Tub-GAL4 to express GAL4 protein in all tissues. In addition, the driver line carried a UAS-linked GFP transgene, which allowed us to use GFP expression for normalization of the mCherry reporter genes (Supplementary Fig. 5).

Expression of mCherry was assessed by fluorescence imaging of formaldehyde fixed salivary glands (SG), central nervous system (CNS) and proventriculus (PV) dissected from otherwise wild type third instar larvae (Fig. 4a). Wild type HA-mCherry expressed well in all imaged tissues. Addition of a polyA track

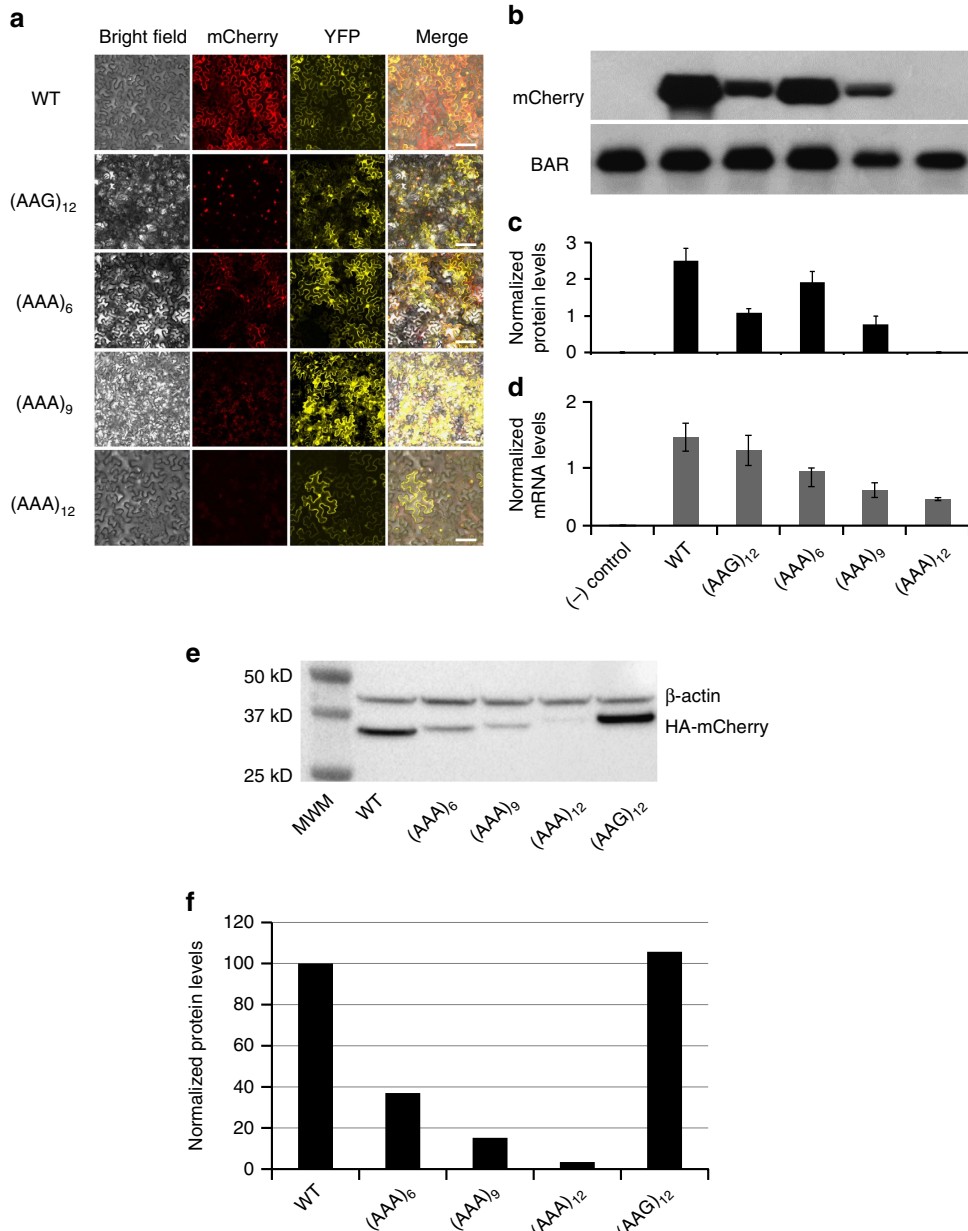

**Figure 3 | Regulation of reporter gene by polyA tracks in the eukaryotic tissue cultures.** (**a**) Fluorescence images of *N. benthamiana* epidermal cells transiently expressing wild type (WT), (AAG)$_{12}$ and (AAA)$_{6-12}$ mCherry constructs. YFP expression was used as a transfection control. The scale bars in images are 100 μm. (**b**)—Western blot analysis, (**c**)—protein level estimate and (**d**)—mRNA levels for transfected ( − ) insert control and WT, (AAG)$_{12}$ and (AAA)$_{6-12}$ mCherry constructs expressed transiently in *N. benthamiana* epidermal cells. (**b**) Primary HA-tag antibody was used for detection of HA-mCherry constructs (molecular weight 34 kDa). Phosphinotricin acetyl transferase (BAR) specific antibody was used as a loading and normalization control (molecular weight 22 kDa) (**c**). Levels of mCherry protein from different constructs were derived from detected band intensities normalized for BAR accumulation detected in the same sample. Error bars represent mean values ± standard error from biological replicates (*n* = 8). (**d**) mRNA levels for different mCherry constructs were calculated as cycle threshold (Ct) values and normalized to *BAR* gene mRNA values. Error bars represent mean values ± s.e. from biological replicates (*n* = 3). (**e**) Western blot analysis of transient mCherry constructs expression in HeLa cells. WT, 12 Lys$_{AAG}$ ((AAG)$_{12}$) and 6–12 Lys$_{AAA}$ ((AAA)$_{6-12}$) mCherry proteins were detected using HA-tag specific primary antibody. β-actin was used as a loading control and was detected using specific antibody. Positions of the fusion protein (HA-mCherry), normalization control (β-actin) and sizes of molecular weight markers (MWM) are indicated. (**f**) Quantification of the mCherry protein levels from detected western blot intensities. Levels of mCherry were normalized to β-actin band intensities and represented as a percentage of the wild type construct values.

with 18A's (6Lys$_{AAA}$) reduced mCherry expression to ∼30% of the wild type construct in all three tissues (Fig. 4b–d; Supplementary Fig. 6). Constructs with 27As and 36As (9 and 12Lys$_{AAA}$ codons, respectively) reduced expression of mCherry in all assayed tissues to ∼20 and 10% of wild type levels, respectively (Fig. 4b–d; Supplementary Fig. 6). Western blot

analyses on cell lysates produced from five fruit fly larvae for each independent construct were consistent with our quantification of fluorescence imaging data (Supplementary Fig. 7). As in the previous experiments with *T. thermophila* and tissue culture systems (Figs 2 and 3), mRNA stability of polyA track constructs in fruit fly larvae exhibited an inverse correlation with the

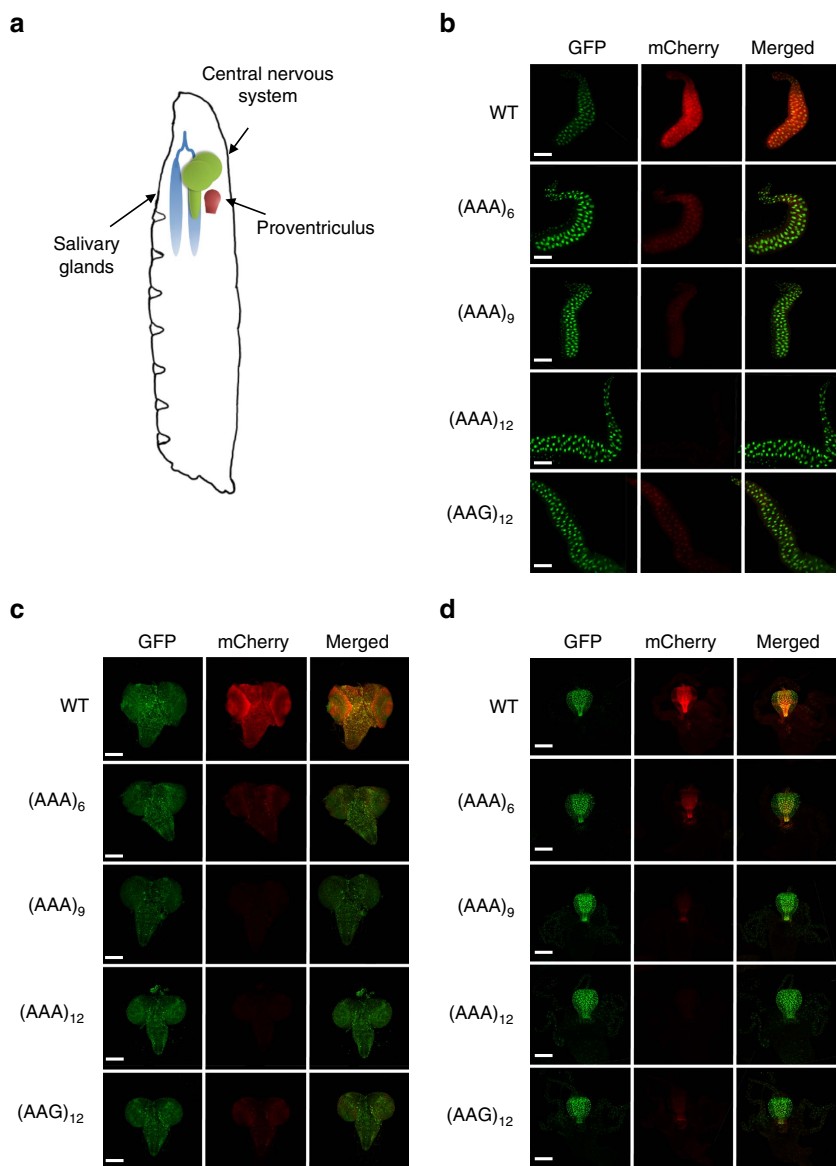

**Figure 4 | PolyA tracks regulate mCherry reporter gene expression in different organs of *D. melanogaster*.** (**a**) Diagram of third instar fruit fly larva showing approximate location of salivary glands (SG, blue), central nervous system (CNS, green) and proventriculus (PV, red). Fluorescence imaging of formaldehyde fixed SG (**b**), CNS (**c**) and PV (**d**), dissected from larvae expressing wild type (WT), (AAG)12 and (AAA)6-12 mCherry constructs. mCherry and GFP indicate images acquired by selective fluorescence filter setting. Overlay of mCherry and GFP fluorescence is shown in the merged panel. The scale bars in images are 200 µm.

length of polyA track (Supplementary Fig. 8) and concordance with protein abundances measured by western blot analyses. Insertion of 12Lys$_{AAG}$ codons had a moderate effect on levels of mCherry mRNA and protein and was similar to the expression levels for the 18As (6Lys$_{AAA}$) insertion construct (Fig. 4b–d; Supplementary Figs 7 and 8). Our data indicate that individual tissues of a complex multicellular organism, such as fruit fly, are equally sensitive to gene expression attenuation mediated by polyA tracks. Therefore, one can use polyA track constructs to create hypomorphic alleles and allelic series, in complex multicellular organisms with similar relative gene expression attenuation efficiencies as observed in the isolated tissues.

One of the potential limitations of polyA tracks could be their homopolymeric nature and observed hypermutability of short tandem repeats (STRs) and homopolymeric sequences that was previously observed in different systems[22,27,28]. To control for the cell population heterogeneity that may arise from hypermutability

of polyA tracks, we have analysed the mutation rate of a single locus insertion of our longest polyA track insertion (36As) in the *D. melanogaster* genome. The sequence of the mCherry reporter gene from genomic DNA isolated from a whole adult fly after more than a year of homozygote crosses, approximately 30 generations of fruit flies, was compared with the original DNA construct that was used for generation of the transgenic insect. Both Sanger and Illumina sequencing of the amplified polyA track (36As) regions from the transgenic fruit fly genome and original DNA vector revealed differences between two data sets (Supplementary Fig. 9). In ~8% of the cases by Illumina sequencing we have observed loss of the polyA track because of the possible recombination event indicated by the similarity of the amplified sequence to the genomic sequence located on the chromosome X and 3L of the fruit fly genome. Illumina sequencing was not reliable enough to show differences in the polyA tracks length because of the difficulty associated with

sequencing of the homopolymers. The Sanger sequencing of the amplified polyA track regions from plasmid and genomic DNA used for the Illumina sequencing revealed a low frequency of mutations in the polyA tracks. We observed insertions as well as polyA track shortening, which is in good concordance with previously published data from different cellular systems[22,27–29]. This is in agreement with our previous analysis of endogenous polyA track genes, which showed rather strong conservation of polyA track sequences in multiple analysed genomes. Taken together, these data indicate that polyA tracks can be used for stable control of gene expression over a multiple generations of model organisms and for an extended period of time without a strong mutational drift in cell mutability.

**PolyA track control is independent of promotor strength**. Our data from the fruit fly experiment indicated that the ratio between reporters with polyA track insertion and control is maintained in all tissues (Fig. 4b–d; Supplementary Figs 6–8). This suggests that inserted polyA tracks maintain their capacity for gene regulation independent of the strength of DNA transcription, which is known to have a large dynamic range across genes and cell types[30].

To systematically evaluate how differences in the strength of transcription would affect gene regulation and hypomorphic expression of reporters with polyA track insertion, we used human Flp-In T-REx 293 cell lines (Thermo Fisher Scientific). Using a protocol for generation of stable and inducible expression cell lines, we generated cells with a single insertion of our HA-mCherry control and HA-$(AAA)_{12}$-mCherry polyA construct in a defined chromosomal locus (Supplementary Fig. 10). The strength of transcription in these cell lines was varied by use of increasing concentrations of doxycycline $(0.001–0.1\,mg\,\mu l^{-1}$ of Dox) added to the growth media, and levels of transcription were assayed in relation to constitutively expressed hygromycin B phosphotransferase (Supplementary Fig. 10). The dose-dependent response of the doxycycline-inducible CMV promoter for both the polyA track and control mCherry transcript ranged over two orders of magnitude. At the same time, relative expression of the polyA track construct was constant; 12–17% expression relative to the control construct based on the western blot analysis (Fig. 5a,b). Moreover, relative mRNA levels of control and polyA track constructs did not change under different transcriptional regimes (Fig. 5c). The steady-state amount of the polyA track construct mRNA was consistently in a range between 1 and 3% of the normalized control construct. The same results are obtained using stable cell lines that express HA-tagged human haemoglobin (delta chain, WT-HBD) and an 18As HBD construct (HBD-6Lys$_{AAA}$) with polyA track inserted in the second exon of the HBD coding sequence (Supplementary Fig. 11). Expression of the HBD-6Lys$_{AAA}$ protein was 2- to 3-fold reduced compared with the WT-HBD construct, based on western blot analysis (Supplementary Fig. 12), and mRNA levels were ~5-fold lower than HBD-WT mRNA levels, measured by qRT-PCR (Supplementary Fig. 13). The relative ratios of WT-HBD and HBD-6Lys$_{AAA}$ protein and mRNA levels were constant for all doxycycline induction levels. Altogether with previous data, showing regulated expression of mCherry reporter in different tissues of the transgenic fruit fly, these data demonstrate that polyA tracks can control gene expression independently of the promoter strength associated with the assayed gene.

**PolyA tracks create hypomorphic mutants in functional genes**. The polyA tracks are mainly composed of lysine residues, AAA or AAG codons, which can be problematic when expressed as

tags because of their charge and propensity for acquiring specific modifications (ubiquitination, acetylation, SUMOylition and hydroxylation). These features of poly-lys chains can potentially influence protein function as well as cell homoeostasis[24,31–33].

We tested our ability to regulate gene expression of functional proteins in both the bacterial and eukaryotic cell systems. In *E. coli*, the chloramphenicol acetyltransferase (*CAT*) gene confers resistance to the broad spectrum antibiotic chloramphenicol (CAM) in a dose-dependent manner[34]. Moreover, the CAT protein is functional only as a homotrimer which has a rather complex protein structure (PDB: 3CLA)[35]. As such, regulation of this gene and subsequent protein folding could be challenging because of the additional lysine residues introduced either by polyA track or control AAG constructs. To show that we can regulate expression of CAT protein by insertion of polyA tracks, we assessed *E. coli* survival under increasing concentrations of CAM in comparison with the wild type *CAT* gene. To control for the influence of additional lysine residues in the N-terminus of the CAT protein, we also inserted 10Lys$_{AAG}$ codons in the N-terminus of *CAT*. Expression of WT-CAT, $(AAG)_{10}$-CAT and $(AAA)_n$-CAT constructs was driven by the inducible arabinose promoter (*pBAD*, Supplementary Fig. 14). All *E. coli* cultures were pulse induced, with addition of 0.1% arabinose, and growth was monitored on LB plates using different concentrations of CAM in the media. WT-CAT and AAG$_{10}$-CAT control constructs were able to survive CAM selection to the same extent (75 mg ml$^{-1}$ CAM, Fig. 6a). Therefore, the function of CAT protein is not affected by the addition of 10 consecutive Lys residues. By contrast, polyA track constructs led to increased CAM sensitivity of *E. coli* cells which correlated nicely with the length of the polyA tracks inserted in the *CAT* gene (Fig. 6a). While the majority of constructs could grow on minimal addition of CAM in the media (15 mg ml$^{-1}$), constructs with 24, 27 or 30As (8, 9 and 10Lys$_{AAA}$) were unable to grow on LB-plates with a CAM concentration of 30 mg ml$^{-1}$. Furthermore, survivability of *E. coli* cells with CAT constructs having 15, 18 and 21As (5, 6 and 7Lys$_{AAA}$) on one hand, and 9 and 12As (3 and 4Lys$_{AAA}$) on the other hand, was impaired when cells were grown on LB-plates with final CAM concentrations of 50 mg ml$^{-1}$ or 75 mg ml$^{-1}$, respectively (Fig. 6a). The survivability of *E. coli* cultures with different CAT constructs was in concordance with expression levels of CAT protein assayed by western blot analyses (Supplementary Fig. 15). The insertion of 10Lys$_{AAG}$ codons in the *CAT* gene did not affect *E. coli* cell growth on CAM selective media or levels of CAT protein expression, arguing that insertion of multiple lysine residues in the N-terminus is not detrimental for the function, structure and stability of CAT protein. These data demonstrate that polyA tracks can regulate levels of certain enzyme expression (CAT) in *E. coli* cells proportionally with their length.

To test the ability of polyA tracks to regulate expression and function of protein in a eukaryotic cell we monitored how polyA tracts affect expression of N-succinyl-5-aminoimidazole-4-carboxamide ribotide synthetase (Ade1, SAICAR) in *S. cerevisiae* (Fig. 6b). The Ade1 protein is a monomeric protein with multiple domains in the structure[36] (PDB: 1A48) and requires strict folding of an active site for catalysis of the seventh step in the purine biosynthesis pathway. Disruption of the *ADE1* gene results in the storage of a red pigment because of the buildup of a metabolic byproduct of the adenine biosynthesis pathway. Colonies of yeast that are *ade1Δ* are a dark red colour; reintroduction of functional Ade1 protein restores the wild type white coloration in a dose-dependent manner. Such differences in colony colour and ability to grown on adenine dropout media (SD-Ade) have been utilized to differentiate strains of yeast prions[37], assess mitotic stability[38] and monitor gene expression[39].

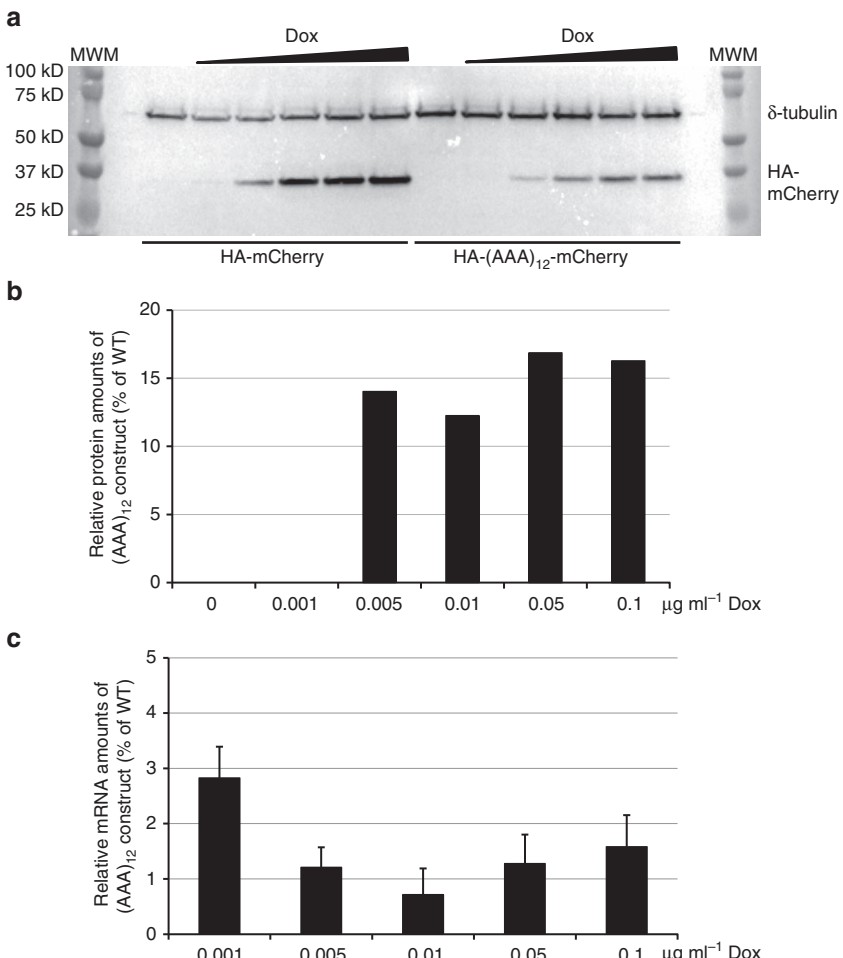

**Figure 5 | PolyA tracks regulate mCherry reporter expression independently of the promoter strength.** (**a**) Western blot analysis of the cell lysates from Flp-In T-REx 293 stable cell lines expressing doxycycline (Dox) inducible wild type (HA-mCherry) and 12 Lys$_{AAA}$ insertion construct (HA-(AAA)$_{12}$-mCherry) from a single locus. Dox concentration in the media was varied from 0 to 0.1 µg ml$^{-1}$. Constitutively expressed δ-tubulin was used as a loading control and was detected using specific antibody. Positions of the fusion protein (HA-mCherry), normalization control (δ-tubulin) and sizes of molecular weight markers (MWM) are indicated. (**b**) Quantification of the mCherry protein levels from detected western blot intensities. Levels of mCherry were normalized to δ-tubulin band intensities and represented as a percentage of the wild type construct values at each Dox concentration. Numbers indicate concentration of Dox in the media. (**c**) Steady state mRNA levels of the 12 Lys$_{AAA}$ insertion construct ((AAA)$_{12}$) measured by qRT-PCR. Relative levels of the mRNA for (AAA)$_{12}$ are presented as percentage of the wild type (WT) construct mRNA level at each Dox concentration. Error bars represent mean ± s.d. values ($n = 3$). Numbers indicate final concentration of Dox in the media.

To survey how polyA tracts affect expression of Ade1, we transformed *ade1Δ* strains of *S. cerevisiae* with single copy plasmids (p416) encoding polyA-*ADE1*-FLAG, with the polyA tracks containing 18, 27 or 36As (6, 9 or 12 Lys$_{AAA}$, Supplementary Fig. 16). Control plasmids contained no insertions (WT) or 12 Lys$_{AAG}$ codons, which showed moderate attenuation of reporter expression in our previous study[17]. Transformants were spotted onto plates to monitor colour phenotypes and growth on media lacking adenine (SD-Ade, Fig. 6b). The empty vector control exhibited a dark red coloration and inability to grow on SD-Ade, consistent with disruption of the *ADE1* locus, while the wild type Ade1-FLAG restored both the white phenotype on media containing adenine (SD-Ura) and growth on SD-Ade. Yeast containing constructs with polyA track lengths of 18, 27 and 36As showed progressively pinker coloration and poorer growth on SD-Ade; however, the control 12 Lys$_{AAG}$ construct conferred a nearly-WT white colour and strong growth on SD-Ade (Fig. 6b). Dot blot analysis of Ade1 protein expression, normalized to total protein, was in accordance with our phenotypic results and revealed visibly reduced amounts of

expression for constructs with insertion of 9 and 12 Lys$_{AAA}$ codons (Supplementary Fig. 16). Expression of Ade1 protein with 12 lysine residues at the N-terminus, as in the case of 12 Lys$_{AAG}$, did not impair folding or function of the assayed protein (Ade1) and exhibited similar phenotypes relative to the insertion of 6 Lys$_{AAA}$ codons. Therefore, addition of polyA tracks to functional genes in both *E. coli* and *S. cerevisiae* preserved protein structure and function but regulated protein abundance. While behaviour of some proteins will of course be sensitive to the inclusion of multiple lysines within their structure, polyA tracks can potentially be quite broadly used in the creation of hypomorphic gene mutants with fixed levels of protein expression.

## Discussion

We have presented a rapid method of generating hypomorphic mutations in a reporter or gene of interest. Insertion of a polyA track into a coding sequence exhibits predictable and robust attenuation of gene expression in all tested cell culture and model

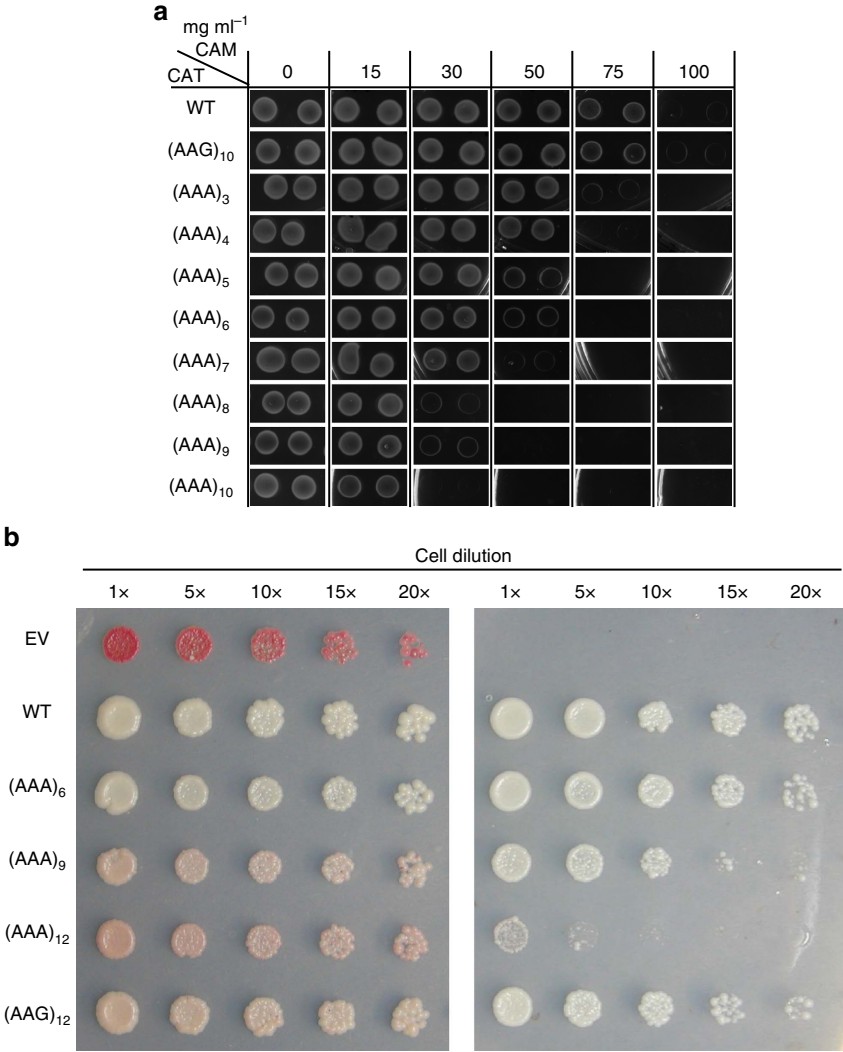

**Figure 6 | Regulation of drug resistance and metabolic survival by insertion of polyA track tags in genes from *E. coli* and *S. cerviseae*. (a)** Survival of *E. coli* cells expressing wildtype (WT), 10Lys$_{AAG}$ ((AAG)$_{10}$) and 3-10 Lys$_{AAA}$ (AAA)$_{3-10}$ chloramphenicol acetyltransferase (CAT) constructs on chloramphenicol (CAM) selective media. Pulse induced *E. coli* cells, expressing different CAT constructs, were plated on selective antibiotic plates with varying amounts of CAM in the media (0–100 mg ml$^{-1}$). Two independent clones were assessed for each construct. *E. coli* colonies were imaged 16 h after plating. **(b)** Assays for *ADE1* gene regulation by polyA tracks ((AAA)$_{6-12}$). Ability of *S. cerevisiae ade1Δ* cells to produce sufficient levels of functional Ade1 protein were assayed by reintroduction of single copy vector with wild type (WT), 12 Lys$_{AAG}$ ((AAG)$_{12}$) and 6–12 Lys$_{AAA}$ ((AAA)$_{6-12}$) Ade1 construct. Empty vector (EV) served as a negative control. Yeast colonies show differential red coloration, on the selective SD-Ura media, which is proportional to the activity of Ade1 protein. Adenine dropout media (SD-Ade) selects for yeast cells expressing sufficient amounts of functional Ade1 protein. Dilutions of the yeast cultures showing relative survival and growth are indicated.

organism systems. The length and the sequence of the polyA track can be manipulated to achieve full range of expression levels, allowing for the generation of an allelic series from the level of a complete knockout to wild type expression for the study of gene function. This method can also be used in synthetic biology applications that require precise gene control and modelling of metabolic and signalling networks[14,15].

The use of polyA tracks overcomes many of the challenges present in current methods of generating hypomorphic mutations and controllable gene expression. For instance, an approach tested for attenuating gene expression in *E. coli* involves mutagenesis of the Shine-Dalgarno sequence (ribosome binding sequence, rbs) in the gene of interest[2,22]. The expression levels from all possible six-mer Shine-Dalgarno sequences were experimentally determined and the information is available in the EMOPEC database[2]. However, this valuable resource would

have to be generated anew to use this approach in other bacteria and it cannot be applied to eukaryotic systems. In addition, many orthogonal translation systems that are used to perturb gene networks rely on modified rbs (refs 40–43). Use of an orthogonal translation system would prohibit use of the rbs for expression regulation. A similar method has overcome this problem by using tandem repeats to change the distance between the rbs and starting ATG codon and, thereby, controlling gene expression levels[22]. However, the restricted use in one system, *E. coli* cells, still limits application of this method. The polyA track system of gene regulation that we describe here for the creation of hypomorphic mutations overcomes these issues because of its dependency on regulation of translation elongation cycle which is well conserved between bacteria and eukaryotes[19].

Hypomorphic mutations have been generated in eukaryotic cell systems by insertion of an antibiotic resistance gene into an

intron[10] or the 3′-untranslated region of a gene[3]. Insertion of the neomycin resistance gene (neo) into an intron introduces a cryptic splice site that results in aberrant splicing of transcripts, effectively reducing gene expression[10]. The reliance on stochastic cryptic splicing events leads to unpredictable changes in transcript expression and is rather gene dependent. Insertion of neo in various genes have resulted in expression of a functionally null allele[44], hypomorphic expression[10,45] or no change in expression[46]. Our system of polyA tracks gives predictable gene expression attenuation in a variety of different eukaryotic systems, and furthermore shows similar relative gene expression attenuation efficiencies in different tissues of the same organism.

We have primarily introduced polyA tracks at the N-terminal regions of reporter genes because of the uniformity of the construct design and to reduce potential frameshifting effects[16,17]. We do not anticipate this to be a major limitation of this method. Our Tetrahymena reporters place the polyA tracts at the N-terminus of YFP, but at the C-terminus of the linked Tetrahymena gene (Supplementary Fig. 3). Furthermore, insertion of polyA track in the second exon of the human beta globin gene (HBD) gene, an unstructured loop of the protein, argue that polyA tracks can be effectively introduced at various positions in the gene (Supplementary Figs 11–13). In addition, we have shown previously that naturally occurring polyA track sequences exist in the human genome and that potential frameshifted products are efficiently degraded by non-sense mediated decay mechanisms[16].

The addition of a polyA track to the target gene will result in additional lysine residues in the protein product. Like any protein tag, it is important to consider the effects of the additional residues when studying the functionality of the protein. We have shown that the function and stability of two structurally diverse proteins, CAT and Ade1, are not affected by up to 12 additional lysine residues. To control for possible effects of the poly-lysine tracks, investigators can create an allele with the same number of lysine residues encoded by AAG codons. The AAG codons will have minimal effect on expression levels while encoding a synonymous protein. Furthermore, the flexibility in polyA track placement within the coding sequence allows investigators to choose the most suitable insertion site for the protein of interest.

The conservation of the polyA track sequences in the multiple genes across vertebrates as well as our analysis of mutation rates of polyA tracks (36As) inserted in the defined locus of D. melanogaster genome argue that polyA tracks can be used to create stable hypomorphic gene alleles. Our results are in the range of already described hypermutability ($\sim$8%) of the short tandem repeats (STRs) and BAT-40 microsatellite (40As) located in the second intron of the 3-beta-hydroxysteroid dehydrogenase gene[28]. The distinction is that our data show general mutation rates for the whole fruit fly after $>$30 generations, while in the case of the mentioned study[28] the mutation rate is dependent on the cell type. An additional study found that the mutation rate in polyA region, 10As in this case, is in the range of $10^{-4}$ per cell per generation[29]. As such, the authors argue that $\sim$1% of cells will be affected by a polyA region mutation in 100 generations. Similar rates were observed in the other studies with $\sim 10^{-6}$–$10^{-2}$ mutation rate for the different lengths of homopolymeric regions or STRs (refs 22,47,48). PolyA tracks used in our study tend to operate on the shorter side of the length distribution of STRs and as such should have similar if not even lower rates of mutations.

PolyA tracks that are used endogenously in eukaryotic genomes are typically interrupted by other nucleotides at various positions within the A-rich sequence, which further reduces potential hypermutability effects. We have observed that the position of the interrupting nucleotide or codon, in combination with the length of the A-rich sequence, modulates gene expression (Supplementary Fig. 2)[16]. These observations suggest that polyA-mediated regulation can be further developed for even more precise control of gene expression. Last, $\sim$2% of human genes are endogenously regulated by polyA tracks, including many well-studied, disease-associated genes, such as BRCA1, TCOF1 and MTDH among others[16,18]. As we showed in our previous study, synonymous mutations of the internal polyA track of such genes can allow investigators to dramatically change expression levels of these genes without manipulation of protein sequence or gene regulatory elements such as promoters and enhancers[16]. The use of our method is not restricted only to these genes, and we feel that the synthetic biology field will benefit from this application. Control of biosynthetic pathways for production of useful secondary metabolites[49], antibiotics[50] or recombinant antibodies[51], as well as introduction of controllable retrosynthetic and fully engineered pathways[52] or ultimate control of metabolic pathways in the modelling of diseases are just a few among the multiple possible applications of this method in the future.

## Methods

**Escherichia coli experiments.** mCherry reporter constructs used for expression in E.coli cells were subcloned using LR clonase recombination (Thermo Fisher Scientific) from pENTR/D-Topo constructs used in this study or in previous studies[16]. The resulting pBAD-DEST49 vector constructs express Thioredoxin (Thrdx) fusion protein as Thrdx-HA tag-polyA-mCherry. For assaying expression of mCherry reporter all constructs were expressed in 2 ml E.coli Top10 strain grown in LB-Carbencilin (LB-Carb; final concentration 100 µg ml$^{-1}$). The cells were grown to optical density at 600 nm (OD$_{600}$) of 0.4 at 37 °C and induced with addition of arabinose (0.5% w v$^{-1}$). Fluorescence of mCherry reporter for each construct was measured in triplicates 2–4 h after induction using Biotek Synergy H4 plate reader (Excitation 475 ± 9, Emission 620 ± 9). The amount of fluorescence was normalized to number of cells measured by OD$_{600}$. To additionally check for expression of fusion proteins, 200 µl of the cells was

**Table 1 | Oligos used for generation of E. coli expressing CAT constructs.**

| Construct/oligo name | Primer sequence |
| --- | --- |
| CAT WT For | CACCATGCACCATCACCATCACCATGAAAAAAAAATCACTGGATATACCACCGTTGATATATCCC |
| CAT 10xAAG | CACCATGCACCATCACCATCACCATGAGAAGAAGAAGAAGAAGAAGAAGAAGAAGAAGATCACTGGATATACCACCGTTGATATATCCC |
| CAT 3xAAA | CACCATGCACCATCACCATCACCATGAAAAAAAAAAAATCACTGGATATACCACCGTTGATATATCCC |
| CAT 4xAAA | CACCATGCACCATCACCATCACCATGAAAAAAAAAAAAAAATCACTGGATATACCACCGTTGATATATCCC |
| CAT 5xAAA | CACCATGCACCATCACCATCACCATGAAAAAAAAAAAAAAAAAATCACTGGATATACCACCGTTGATATATCCC |
| CAT 6xAAA | CACCATGCACCATCACCATCACCATGAAAAAAAAAAAAAAAAAAAAATCACTGGATATACCACCGTTGATATATCCC |
| CAT 7xAAA | CACCATGCACCATCACCATCACCATGAAAAAAAAAAAAAAAAAAAAAAAATCACTGGATATACCACCGTTGATATATCCC |
| CAT 8xAAA | CACCATGCACCATCACCATCACCATGAAAAAAAAAAAAAAAAAAAAAAAAAAATCACTGGATATACCACCGTTGATATATCCC |
| CAT 9xAAA | CACCATGCACCATCACCATCACCATGAAAAAAAAAAAAAAAAAAAAAAAAAAAAAATCACTGGATATACCACCGTTGATATATCCC |
| CAT 10xAAA | CACCATGCACCATCACCATCACCATGAAAAAAAAAAAAAAAAAAAAAAAAAAAAAAAAATCACTGGATATACCACCGTTGATATATCCC |
| CAT Rev | CATTACAGATCTTCTTCAGAAATAAGTTTTTGTTCCGCCCCGCCCTGCCACTCATCGCAG |

harvested 2 h post-induction, resuspended in 100 μl of 2 × SDS sample buffer and analysed by SDS–polyacrylamide gel electrophoresis (SDS–PAGE) followed by western blot analysis using HA-tag specific probe (Santa Cruz Biotechnology Inc.). Images of western blot analyses were generated by Bio-Rad Molecular Imager ChemiDoc XRS System with Image Lab software for chemi-luminescence detection (Supplementary Fig. 17).

The chloramphenicol acetyltransferase (CAT) constructs used for functional protein studies were created by amplification of the CAT gene from pENTR/D-Topo vector (Thermo Fisher Scientific) using primers listed in Table 1. Constructs were subcloned into pBAD-DEST49 vector for use in functional assays. E. coli Top10 cells freshly transformed with pBAD-DEST49 plasmids expressing CAT reporters with different polyA tracks, as well as CAT control reporters were grown in liquid LB-Carb media (100 μg ml$^{-1}$). For the chloramphenicol (CAM) survivability assay E. coli cells were grown to $OD_{600} = 0.4$ and non-induced fractions were spotted on LB-Carb plates (Carb 100 μg ml$^{-1}$) without chloramphenicol. The residual amount of the cells was induced for 1 h with arabinose (final concentration 0.1% (w v$^{-1}$)). Fraction of cells, 100 μl, was harvested after 30 min of induction, resuspended in 50 μl of 2 × SDS sample buffer and analysed by SDS–PAGE followed by western blot analysis using HA-tag specific probe (Santa Cruz Biotechnology Inc., SC-7392HRP). Reminder of cells were washed twice in M9 minimal media, resuspended in the starting volume of LB-Carb media and 5 μl of cells was spotted as induced fraction on LB-Carb or to LB-Carb/CAM plates with raising amount of chloramphenicol in the media (CAM 15–100 μg ml$^{-1}$). Plates were incubated overnight at 37 °C and imaged 24 h post-induction using Bio-Rad Molecular Imager ChemiDoc XRS System (Supplementary Fig. 21).

**Saccharomyces cerevisiae experiments.** To conduct functional studies with polyA track-induced hypomorphic attenuation in S. cerevisiae cells the ADE1 locus was deleted from 74-D964 yeast strain via homologous recombination. Resultant ade1Δ strains (Table 2) were transformed with an empty vector or a plasmid-based ADE1 containing variable length of polyA tracks, as well as WT and 12Lys$_{AAG}$ insertion constructs. Constructs were generated by performing PCR on ADE1 isolated from the yeast genomic tiling library (Open Biosystems) with primers listed in Table 3. PCR products were digested and ligated into a p416 vector backbone containing the ADE1 endogenous promoter. Clones were verified via sequencing and correct constructs were transformed into ade1Δ deletions strains via the PEG-LiOAc method (Table 2). To generate dilution spottings, three colonies were picked from each transformation plate and grown overnight in selective media. In the morning, cultures were normalized to $OD_{600} = 1.0$ and 10 μl of cells were spotted onto rich media, SD-Ura and SD-Ade for phenotypic analysis.

Relative protein abundance was determined via dot blot. Briefly, yeast transformants were picked from selection plates to inoculate 10 ml of SD-Ura and grown overnight to ~OD = 0.6. In the morning, cells were harvested and lysed in buffer (25 mM Tris–HCl pH 7.5, 50 mM KCl, 1 mM EDTA, Roche protease inhibitor cocktail) via mechanical disruption with acid-washed glass beads (Sigma). Total protein was normalized to 1 mg ml$^{-1}$ via Bradford assay, and 20 μg of total protein was spotted onto a nitrocellulose membrane. Western blotting was performed by overnight incubation with anti-Flag (Sigma M2, F3165, 1:1,000 in 5% milk) and goat anti-rabbit (Sigma A0545, 1:10,000 in 5% milk) antibodies, followed by detection with chemiluminescence (Amersham ECL).

**Tetrahymena thermophila experiments.** T. thermophila strain B2086 (II) was used for all experiments reported. Similar results were obtained with strain CU428 ((VII) mpr1-1/mpr1-1). To assess the effect of polyA-tracks on protein accumulation, we modified a fluorescent protein tagging vector, pBSICY-gtw (ref. 53) so as to fuse YFP to the carboxyl-terminus of a Macronucleus-Localized Protein 1 of unknown function (MLP1, TTHERM_00384860), separated by a Gateway recombination cassette (Invitrogen/Life Technologies, Inc) and expressed from the cadmium inducible MTT1 promoter[54]. The MLP1 gene coding region was amplified with oligonucleotides 5′ ALM Bsi' 5′–CAC CCG TAC GAA TAA AAT GAG CAT TAA TAA AGA AGA AGT–3′ and 3′ ALM RV 5′–GAT ATC TTC AAT TTT AAT TTT TCT TCG AAG TTG C 3′ and cloned into pENTR-D in a topoisomerase mediated reaction before digesting with BsiWI and EcoRV and inserting into BsiWI/PmeI digested pBSICY-gtw. Subsequently, LR Clonase II was used to insert a linker containing the sequence coding for an HA epitope tag alone (WT) or the tag plus different length of polyA tracks or AAG insertions in place of the Gateway cassette.

The expression cassette is located within the 5′ flanking region of a cycloheximide resistant allele of the rpL29 gene to direct its integration into this genomic locus. These constructs were linearized with PvuI and SacI in the region flanking the

Tetrahymena rpl29 sequences and introduced into starved Tetrahymena cells by biolistic transformation[55,56]. Transformants were selected in 1x SPP medium containing 12.5 μg ml$^{-1}$ cycloheximide. To control for copy number, PCR assays with primers MTT2386 5′–TCTTAGCTACGTGATTCACG–3′ and Chx-117, 5′- ATGTGTTATTAATCGATTGAT–3′ and Chx85r, 5′–TCTCTTTCATGCATGC TAGC–3′ verified that all rpL29 loci contained the integrated expression construct.

Transgene expression was induced by addition of 0.6 μg ml$^{-1}$ CdCl$_2$ and cells were grown 12–16 h before monitoring protein accumulation. YFP accumulation was visualized by epifluorescence microscopy as previously described[57]. Whole cells extracts were generated by boiling concentrated cell pellets ($2 \times 10^6$) in 120 μl of 1 × Laemmli sample buffer, followed by fractionation on 10% SDS polyacrylamide gels and transfer to nitrocellulose. YFP accumulation was a monitored with mouse anti-GFP antisera (G28R anti-GFP (OAEA00007) antibody, Aviva Systems Biology) and normalized to acetylated Rabbit anti-Histone H3 trimethyl-lysine (Upstate Biotechnologies/Millipore, NY, 07-473). qPCR analysis was done using 5′–AGGCCTACAAGACCAAGGGT–3′ and 5′–AGAGCGGTTTT GACGTTGGA–3′ primers for T. thermophila ribosomal protein L21 (rpl21) which was used for normalization. Primers 5′–CCCGTATGACGTACCGGATTATG–3′ and 5′–ACTTCAGGGTCAGCTTGCC–3′ were used for detection and estimation of fusion protein transcript levels using SybrGreen master mix and CFX96 Touch Real time PCR Detection System (BioRad). Normalized $\Delta Ct$ values were used to calculate fold ratio between WT, 12Lys$_{AAG}$ and polyA track constructs.

**Nicotiana benthamiana experiments.** Constructs for expression of HA-tagged mCherry reporters that were already cloned in pEntryD-TOPO vector were sub-cloned to pEarleyGate 100 (ABRC stock number CD3-724) through LR reaction using LR clonase (Invitrogen). The mCherry reporter constructs, pEARLY100 and pBIN61 plasmids were individually electroporated into Agrobacterium tumefaciens strains GV3101 (ref. 58). The strain carrying pBIN61 construct expressing p19 protein from tomato bushy stunt virus was co-infiltrated with the reporter constructs to suppress post-transcriptional gene silencing[59]. The Agrobacterium suspensions carrying the reporter constructs were infiltrated into the leaves of 5- to 6-week-old N. benthamiana plants as described in Joensuu et al., 2010 (ref. 60). Briefly, saturated over-night cultures were spun-down and resuspended in the infiltration solution (3.2 g l$^{-1}$ Gamborg's B5 plus vitamins, 20 g l$^{-1}$ sucrose, 10 mM MES pH 5.6, 200 μM 4′-Hydroxy-3′,5′-dimethoxyacetophenone) to a final $OD_{600} = 1.0$; Agrobacterium suspensions carrying the reporter constructs were individually mixed with suspensions carrying the pBIN61 construct in 1:1 ratio before infiltrations. These suspensions were infiltrated into separate segments of two young leaves on each of eight different N. benthamiana plants, which served as biological replicates. For control, 1:1 suspension of A. tumefaciens carrying pEARLY 100 with no insert along with pBIN61 was used. The infiltrated plants were maintained in a controlled growth chamber conditions at 22 °C, with a 16 h photoperiod.

Samples of the abaxial epidermis of N. benthamiana leaves infiltrated with different mCherry reporter constructs were collected 6 days post-infiltration. Infiltration was performed as described in the previous section, with the addition of an YFP-expressing construct pEARLY104 (ABRC stock number CD3-686), which served as infiltration control. The samples were visualized for fluorescence by confocal laser-scanning microscopy using a Leica TCS SP2 confocal microscope. Samples for RNA and total soluble protein (TSP) extraction were separately collected from the infiltrated plants 6 days post-infiltration using a cork borer (7.1 mm in diameter); each sample contained equal amounts of leaf tissue (TWO leaf discs) collected from each of the segments on the two leaves infiltrated with the same construct.

Analysis of mCherry protein accumulation was carried out by western blot as described in Gutiérrez et al. and Conley et al.[61,62] Briefly, phosphate-buffered saline (PBS: 8 g l$^{-1}$ NaCl, 1.16 g l$^{-1}$ Na$_2$HPO$_4$, 0.2 g l$^{-1}$ KH$_2$PO$_4$, 0.2 g l$^{-1}$ KCl, pH 7.4), supplemented with 1 mM EDTA, 1 mM phenylmethanesulfonylfluoride (PMSF), 1 μg ml$^{-1}$ leupeptin 0.1% Tween-20 and 100 mM sodium L-ascorbate was used for total soluble protein (TSP) extractions. Bradford assay (Biorad) was used to quantify TSP in the extracts using a standard curve ($r^2 = 0.99$) of known concentrations of Bovine Serum Albumin (BSA). Sample extracts (25 μg TSP for mCherry and 5 μg TSP for phosphinotricin acetyl transferase (BAR) protein detection) were separated by SDS–PAGE, blotted onto nitrocellulose membrane and probed with a primary anti-HA tag antibody (Genscript, A01244-100) for mCherry or anti-Phosphinotricin acetyl transferase antibody (Abcam, Ab50504) for BAR, both at 1:2,000 dilution, followed by HRP-conjugated secondary antibody (Biorad, 170-6516 and 170-6515) at 1:5,000 dilution. The blots were washed (3 times × 10 min) in 1 × Tris-buffered Saline (TBS, 50 mM Tris, 150 mM NaCl, pH 7.5) containing 0.1% Tween (Sigma) and images were obtained after 1 min

### Table 2 | S. cerevisiae strains used in this study.

| Name | Strain background | Genotype |
| --- | --- | --- |
| HT 971 | 74-D694 | Mat A, ade1::KANMX4, trp1-289(UAG), his3Δ-200, ura3-52, leu2-3, 112 |
| HT 972 | 74-D694 | Mat A, ade1::KANMX4, trp1-289(UAG), his3Δ-200, ura3-52, leu2-3, 112 |

**Table 3 | Primers used for generation of ADE1 constructs.**

| Name | Sequence |
| --- | --- |
| FwdAde1SpeIWT | 5′GGactagtATGTCAATTACGAAGACTGAACTGG |
| FwdAde1SpeI6AAA | 5′GGactagtATGAAAAAAAAAAAAAAAAAATCAATTACGAAGACTGAACTGG |
| FwdAde1SpeI9AAA | 5′GGactagtATGAAAAAAAAAAAAAAAAAAAAAAAAAATCAATTACGAAGACTGAACTGG |
| FwdAde1SpeI12AAA | 5′GGactagtATGAAAAAAAAAAAAAAAAAAAAAAAAAAAAAAAAAATCAATTACGAAGACTGAACTGG |
| FwdAde1SpeI12AAG | 5′GGactagtATGAAGAAGAAGAAGAAGAAGAAGAAGAAGAAGAAGAAGTCAATTACGAAGACTGAACTGG |
| RevAde1ClaIFLAG | 5′GGatcgatTTACTTGTCGTCATCGTCCTTGTAGTCGTGAGACCATTTAGACCC |
| FwdAdePromSacI | 5′GGgagctcACACGATAGCAAAGCAG |
| RevAdePromXbaI | 5′ GGtctagaTATCGTTAATATTTCG |

incubation with the enhanced chemiluminescence detection system (GE Healthcare) (Supplementary Fig. 18). Numerical values for protein accumulation were derived from the detected band intensities on the analysed images using TotalLab TL 100 software (Nonlinear Dynamics, Durham, USA). The mCherry accumulation values were normalized for BAR accumulation detected in the same sample. Normalized values of the mCherry protein accumulation for each reporter construct were presented as the mean of eight biological replicates ± s.e.; Tukey's honest significance test (JMP software, SAS Institute Inc.) was used to identify significantly different means ($\alpha = 0.05$).

For quantitative RT–PCR (qPCR), total RNA was extracted using an RNeasy plant mini kit coupled with DNase treatment (Qiagen). The purified RNA (500 ng) was reverse-transcribed using the Maxima first-strand complementary DNA (cDNA) synthesis kit (Thermo Fisher Scientific). The resulting cDNA ($2 \, ng \, \mu l^{-1}$) was quantified by qPCR using the Maxima SYBR Green/ROX qPCR master mix (Thermo Fisher Scientific) and CFX384 Touch Real-Time PCR Detection System (Biorad). Cycle threshold (Ct) values were normalized to phosphinothricin N-acetyltransferase (BAR) gene expressed from the same plasmid used for transient expression. Primer sequences used: For mCherry—mCherryFWD: 5′–GGCTAC CCATACGATGTTCC–3′; mCherryREV: 5′–CCTCCATGTGCACCTTGAAG–3′; for BASTA–BAR-F3: 5′–TCAAGAGCGTGGTCGCTG–3′ and BAR-R3: 5′–CAA ATCTCGGTGACGGGCAG–3′.

**Drosophila melanogaster experiments.** Reporter gene expression was achieved with the GAL4/UAS system. The UAS-mCherry transgene plasmids were constructed from the phiC31 integrase plasmid, pJFRC28-10XUAS-IVS-GFP-p10 (Addgene plasmid #36431)[63]. GFP was removed by digestion with KpnI and XbaI and replaced with HA-mCherry and HA-polyA-mCherry. Transgenic flies were obtained by injecting P{CaryP}attP2 embryos with each pJFRC28 mCherry construct to achieve site-specific, single insertion on the third chromosome at the attP2 landing site (Rainbow Transgenic Flies, Inc.). Injected $G_0$ adult flies were backcrossed to $w^{1118}$ flies. Red-eyed progeny indicated successful germline integration of the UAS-mCherry expression cassette. Male red-eye progeny were crossed to female w;TM3 Sb/TM6 Tb flies followed by sib-crosses of the $F_1$ progeny to generate homozygous UAS-mCherry transgenic lines. Insertion was confirmed by Sanger sequencing of PCR amplified mCherry from genomic DNA of individual flies. Each mCherry transgenic fly line was crossed to a TubGal4 UAS-GFP driver line (derived from BSC42734) to achieve mCherry expression in all tissues. GFP expression was used for normalization. All flies were maintained at 25 °C.

Third instar larvae from each cross were fixed in formaldehyde and dissected to recover the salivary glands (SG), intact central nervous system (CNS) and proventriculus (PV). The tissues were mounted on glass cover slips and confocal images were taken on a Zeiss Imager 2 upright microscope using identical parameters for all images of each tissue type. Fluorescence intensity of the mCherry and GFP were quantified with Zen 9 software.

Total RNA was extracted from each cross by pooling five third instar larvae in 1.5 ml RNase-free Eppendorf tubes which were then frozen in dry ice. Frozen samples were homogenized using 1.5 ml pestles (Fisherbrand, RNase- and DNase-free). After homogenization, 1 ml RiboZol reagent (Amresco) was added and extraction was completed according to manufacturer's instructions. Total RNA samples were treated with TURBO DNA-free kit (Ambion) to remove potential genomic DNA. cDNA synthesis was performed with iScript Reverse Transcription Supermix (Bio-Rad) with 1 µg of total RNA in a 20 µl reaction. RT-qPCR was performed in the Bio-Rad CFX96 Real-Time System with iQ SYBR Green Supermix (Bio-Rad). The mCherry transcript was detected with the following primers: 5′–TGACGTACCGGATTATGCAA–3′ and 5′–ATATGAACTGAGGGG ACAGG–3′. Cycle threshold (Ct) values were normalized to EF1 with the following primers: 5′–GCGTGGGTTTGTGATCAGTT–3′ and 5′–GATCTTCTCCTTGCCC ATCC–3′[64].

For western blot analysis, five third instar larvae from each cross were collected and frozen in dry ice. Frozen samples were homogenized using 1.5 ml pestles (Fisherbrand, RNase- and DNase-free). After homogenization, SDS sample buffer was added and the samples were boiled for 10 min. Anti-HA tag (Santa Cruz Biotechnology Inc., SC7392HRP) was used to detect mCherry expression. Samples were normalized with anti-GFP (Clontech, 632381).

For Illumina sequencing analysis, genomic DNA was isolated from a single adult HA-(AAA)12-mCherry fly approximately 30 generations after transgene insertion. The QIAGEN DNeasy Blood and Tissue Kit were used, following the supplementary protocol for total DNA extraction from insect cells. The polyA track of the mCherry transgene was amplified from both the genomic DNA and the plasmid DNA that was used to generate the transgenic fly line. The primers (5′–GGCTGCTCGAGCCGTATGACGTACCGGATTATGC–3′ and 5′–CCT TGTGATCAGGCCATGTTATCCTCCTCGC–3′) added XhoI and SpeI restriction enzyme sites flanking the polyA track region. PCR products were purified with NucleoSpin Gel and PCR Clean-up (Macherey-Nagel) and digested with XhoI and SpeI to generate overhangs for ligation of Illuminia adaptor sequences. To enrich for amplicons with both adaptor sequences, the product was amplified with the following primers: 5′- -3′ and 5′- -3′. Samples were run as a spike-in on an Illumina HISeq machine. Reads that matched the first 32 or 33 expected nucleotides, depending on the indexing barcode used, were counted. These resulted in ~670,000 genomic reads and 1,000,000 plasmid reads after removing sequences that had fewer than 50 reads.

For Sanger sequencing analysis, the polyA track from the genomic DNA extracted for Illumina sequencing and the original plasmid were amplified with the following primers: 5′-GGCTGCTCGAGCCGTATGACGTACCGG-3′ and 5′-ACA TGAACTGAGGGGACAGG-3′. The PCR products were purified with NucleoSpin Gel and PCR Clean-up kit (Macherey-Nagel) and ligated into the pCR-Blunt vector using the Zero Blunt PCR Cloning Kit (Thermofisher). The vector was transformed into Max Efficiency DH5α competent cells (Thermofisher) and 60 colonies from both the genomic DNA and plasmid DNA were sent for sequencing (Genewiz). Sequences with good quality scores were used for downstream analysis.

**Homo sapiens cell culture experiments.** mCherry reporter constructs used for transient expression in human cells were subcloned using LR clonase recombination (Thermo Fisher Scientific) from pEntryD-Topo constructs used in other experiments or in previous studies[16]. DNA fragments for constructs used for creation of inducible and stable cell lines were PCR amplified, purified and ligated into pcDNA 5/FRT/TO vector (Thermo Fisher Scientific).

HeLa cells were cultured in Dulbecco's modified Eagle's medium (DMEM) (Gibco) and supplemented with 10% fetal bovine serum, 5% minimum essential medium nonessential amino acids ($100 \times$, Gibco), 5% penicillin and streptomycin (Gibco) and L-glutamine (Gibco). Flp-In T-REx 293 cells were grown in the same media with addition of $5 \, \mu g \, ml^{-1}$ of blasticidin and $100 \, \mu g \, ml^{-1}$ of Zeocin for non-recombined cells, or $5 \, \mu g \, ml^{-1}$ of blasticidin and $100 \, \mu g \, ml^{-1}$ of hygromycin for growth of stable cell lines expressing mCherry or HBD constructs.

Plasmids were introduced to the tissue culture cells by the Neon Transfection System (Thermo Fisher Scientific) using 100-µl tips according to cell-specific protocols (www.lifetechnologies.com/us/en/home/life-science/cell-culture/transfection/transfection---selection-misc/neon-transfection-system/neon-protocols-cell-line-data.html). Hela cells, used for transient expression, were electroporated with 1.5 µg of DNA plasmids and were harvested 24 h after the electroporation. Flp-In T-REx 293 cells were electroporated with plasmids, selected for positive clones as described by protocol (https://tools.thermofisher.com/content/sfs/manuals/flpinsystem_man.pdf). Expression of polyA track and control constructs was induced by addition of various amounts of doxycycline from a common stock ($1 \, mg \, ml^{-1}$) and harvested 24 or 48 h after induction, if not indicated differently.

Total RNA was extracted from cells using the RiboZol RNA extraction reagent (Amresco) according to the manufacturer's instructions or using GenEluteTM Direct. RiboZol reagent (500 µl) was used in each well of 6- or 12-well plates for RNA extraction. Precipitated nucleic acids were treated with Turbo deoxyribonuclease (Ambion), and total RNA was dissolved in ribonuclease-free water and stored at −20 °C. RNA concentration was measured by NanoDrop (OD260/280). iScript Reverse Transcription Supermix (Bio-Rad) was used with 1 µg of total RNA following the manufacturer's protocol. RT-qPCR was performed in the Bio-Rad CFX96 Real-Time System with iQ SYBR Green Supermix (Bio-Rad). For both transient expression samples and stable cell line samples, the mCherry transcript was detected with the following primers: 5′-TGACGTACCG GATTATGCAA-3′ and 5′-ATATGAACTGAGGGGACAGG-3′. Cycle threshold (Ct) values were normalized to the neomycin resistance gene expressed from the

same plasmid for transient expression (5′-CTGAATGAACTGCAGGACGA-3′ and 5′-ATACTTTCTCGGCAGGAGCA-3′) or hygromycin (5′-GATGTAGGAGGGC GTGGGATA-3′ and 5′-ATAGGTCAGGCTCTCGCTGA-3′ or actin gene for stable cell lines (5′-AGAAAATCTGGCACCACACC-3′ and 5′-AGAGGCGTACAGGGA TAGCA-3′.

Total cell lysates were prepared with passive lysis buffer (Promega). Blots were blocked with 5% milk in 1× tris-buffered saline with 0.1% Tween 20 (TBST) for 1 h. Horseradish peroxidase–conjugated or primary antibodies (anti-β-actin (Santa Cruz Biotechnology Inc., SC1616HRP), anti δ-tubulin (Sigma-Aldrich, T3950)) were diluted according to the manufacturer's recommendations and incubated overnight with the membranes. The membranes were washed four times for 5 min in TBST and prepared for imaging, or secondary antibody (Cell Signaling Technology, 7076S) was added for additional 1 h incubation. Images were generated by Bio-Rad Molecular Imager ChemiDoc XRS System with Image Lab software by chemiluminescence detection or by the LI-COR Odyssey Infrared Imaging System (Supplementary Figs 19 and 20). Blots imaged by the LI-COR system were first incubated for 1 h with Pierce DyLight secondary antibodies.

**Data availability.** The data that support the findings of this study are available from the authors on reasonable request; see author contributions for specific data sets.

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

## Acknowledgements

We thank D. Piston, M. Jovanovic, T. Schedl and P. Szczęsny and members of Djuranovic lab for helpful comments. This work was supported by the NIH grant T32 GM007067 to LLA and KMK, National Science Foundation grant # 1412336 (MCB, PI: Chalker) to JCC and DLC, the NSERC-DG to VG, NIH Grant R01 NS036570 to JBS, NIH Grant R01 GM072778 to HT, NIH Grant R01 GM112824 and the American Cancer Society (grant IRG-58-010-58-2) to SD. Subject to the U.S. Provisional Patent Application, Serial No. 62/361,307.

## Author contributions

R.G. and S.D. are responsible for the initial idea and design of the study. S.D. and S.P.D. generated and validated initial expression constructs. S.D. and L.L.A. performed experiments in *E. coli* and human cell cultures. L.L.A. performed experiments in *D. melanogaster* with the guidance of J.B.S. J.J.C. and D.L.C. conducted experiments in *T. thermophila*. P.J., V.G., I.K. and R.M. designed experiments in *N. benthamiana*; P.J. and I.K. completed the experiments. K.M.K. and H.L.T. performed experiments in *S. cerevisiae*. All authors contributed to writing the manuscript.
