## [Peer Review File · Nature Communications]

Reviewers' Comments:

Reviewer #1 (Remarks to the Author)

Previous work by this team have shown that polyA tracks inhibit translation and decrease the stability of mRNA in both eukaryotes and bacteria (ref 16 & 17). In this paper the team demonstrate that the insertion of polyA tracks into the coding sequences of reporter and functional genes can be used to modulate expression levels across a wider range of organisms. They show that expression levels decrease as the length of the polyA track increases and that this is independent of the promoter strength. Overall I agree with the authors that this is a simple and useful method that could have numerous applications in gene functional analysis and synthetic biology. The fact that it could be applied across eukaryotic and prokaryotic organisms makes it a particularly attractive strategy compared with other approaches which are typically more organism specific.

The only problem with the approach is that the proteins produced have up to 12 additional Lys residues. Whilst these residues did not effect the function of the few gene products described in the paper, this would not necessarily be the case for all genes of interest. The introduction of N-terminal poly-lysine residues, for example, could effect protein folding, stability, protein-protein interactions or impair catalytic function etc., in the same way that His-tags and other protein fusions have been shown to effect the function of some proteins. Consequently for genes of unknown function, or gene products with activity that was difficult to assay, there would always be some uncertainty whether the effects one observed in vivo was due to reduced expression level or impaired protein functionality.

Finally the paper suggests (line 121-122) that the authors hypothesized that expression levels decrease with increasing polyA track length. However the two earlier papers (ref 16 & 17) already showed that this is the case in both eukaryotes and bacteria. Thus whilst this current paper does provide a more extensive study across a wider range of organisms, some of its main conclusions had already been reached in the two earlier publications. The paper/project could be strengthened by demonstrating a real application in gene functional analysis or synthetic biology, in addition to measuring expression levels of a few well know genes that are easy to quantify/analyse.

Reviewer #2 (Remarks to the Author)

Overall assessment

This novel, well-written manuscript describes the use of polynucleotide repeat tracts within gene coding regions for a range of cross-kingdom hosts to fine-tune gene expression through translational attenuation. I applaud the creative utilization of repeat tracts in this manner and am impressed to see its general applicability across a broad range of organisms.

I particularly like the concept of modifying existing genes that have poly-A tracts to fine-tune expression or to introduce these tracts with or without repeat degeneracy to introduce a method to control expression. These ideas are certainly a topic of interest to synthetic biologists and could be stressed more clearly in the introduction and discussion, especially for model organisms that currently lack reliable systems to tune gene expression.

I do, however, have reservations about the reliability and reproducibility of the approach given the high rate of insertion-deletion mutations documented for long poly-nucleotide repeats (Natural variation: Brinkmann et al AJHG 1998, Ellegren Nat Rev Genet 2004; engineering / synthetic biology context: Egbert & Klavins, PNAS 2012). In addition to citing relevant work on repeat instability, I suggest the authors address the following items to rework the manuscript for

publication.

My main concern that is not addressed in the manuscript is the population heterogeneity that is certainly introduced by long repeat elements that are subject to high-rate insertion/deletion mutations. While the discussion points out some challenges with existing expression optimization systems but fails to identify limitations of the authors' approach, specifically with regard to the hypermutable nature of long mono-nucleotide repeat tracts. Not addressing this concern could severely limit the impact of the authors' approach.

Many applications may require long-term expression of hypomorphic target genes over many generations, suggesting that mutations in a hypervariable poly-A tract could generate significant cell-cell genetic variability, limiting quantitative assessment of gene expression either spatially or temporally.

While the ultimate destination of this approach in most cases is single-copy chromosomal integration, many of the test conditions for the manuscript are transient transfections or plasmid transformations, each of which provide a population of target genes to individual cells. Multi-copy gene dosage may mask the effect of mutagenesis on repeat-modulated gene expression.

The expression attenuator design utilizes the poly-A tract either as an N-terminus tag or between two components of a fusion protein. In the case of the fusion protein, an insertion/deletion mutation to the repeat element may eliminate fluorescence but the N-terminus protein would remain functional. Thus, a fluorescent readout of attenuation does not necessarily indicate expression.

I would like to see the authors address this concern by quantifying the mutation rate for one or more of the test systems. One simple test could be to perform flow cytometry of fluorescently labeled E. coli to identify what fraction of the population is non-fluorescent as a function of cell generation compared to controls, though this may require genomic integration to quantify if the plasmid copy number, and thus masking effect, is high.

Secondary questions

Line 348 (discussion): how is this method necessarily rapid? It seems that simple, rather than rapid, better describes the approach. I agree that simple, predictable mapping of sequence to expression may allow rapid pinpointing of optimal expression levels. The manuscript could be more clear in assessing the uniqueness of the approach.

Have the authors attempted to use of other repeat elements to attain greater control over gene expression? Poly-T or poly-AT tracts, for instance? Some treatment of the feasibility of extensions with other repeats in the discussion could help frame the outlook for finer control or additional applications.

For figure 2d-f, the (AAG)₁₂ values for protein and mRNA levels are not consistent. The protein band should be 60% of WT based on qPCR data, yet it hardly appears on the western blot. Panel F appears to be the outlier, while D & E appear consistent. The inconsistency should be addressed in the text or the experiments redone for clarify.

In figure 3, the results for (AAG)₁₂ in panels c and d again appear to be inconsistent. The qPCR results appear similar to wild-type but the protein levels are significantly lower.

The text addresses these concerns in part in lines 219-222, but this seems buried or overlooked. The authors could consider reworking the text or expanding the treatment of the apparent inconsistencies to help clarify the results.

Miscellany

Line 261 - seems it should reference Figure 4, not Figure 3.

Line 271 concentration listed for Dox is not consistent with Figure 5 or Figure 5 caption

Line 275 - Relative protein for Figure 5b appears 12-17% WT, inconsistent with text

Figure 5 caption, panel b - it could be more clear here whether the percent WT measure in (b)/(c) is calculated for each Dox level or a single value (such as at max Dox) to normalize all samples. I assume the former to show a consistent fraction of expression at different induction levels, but some more textual clarity would help.

Lines 563-569 - do "dox concentration" for (b) and "final dox concentration" for (c) mean the same thing here? If so, unify the text descriptions.

Supplement line 93 - "was monitored"?

Caption for Supp fig 7 - standard deviation at end of caption appears truncated to "standard"

Reviewer #1 (Remarks to the Author):

Previous work by this team have shown that polyA tracks inhibit translation and decrease the stability of mRNA in both eukaryotes and bacteria (ref 16 & 17). In this paper the team demonstrate that the insertion of polyA tracks into the coding sequences of reporter and functional genes can be used to modulate expression levels across a wider range of organisms. They show that expression levels decrease as the length of the polyA track increases and that this is independent of the promoter strength. Overall I agree with the authors that this is a simple and useful method that could have numerous applications in gene functional analysis and synthetic biology. The fact that it could be applied across eukaryotic and prokaryotic organisms makes it a particularly attractive strategy compared with other approaches which are typically more organism specific.

The only problem with the approach is that the proteins produced have up to 12 additional Lys residues. Whilst these residues did not effect the function of the few gene products described in the paper, this would not necessarily be the case for all genes of interest. The introduction of N-terminal poly-lysine residues, for example, could effect protein folding, stability, protein-protein interactions or impair catalytic function etc., in the same way that His-tags and other protein fusions have been shown to effect the function of some proteins. Consequently for genes of unknown function, or gene products with activity that was difficult to assay, there would always be some uncertainty whether the effects one observed in vivo was due to reduced expression level or impaired protein functionality.

Author's response:

We are thankful to the reviewer for his overall positive attitude and comments on simplicity, usefulness and applicability of our method on the broad set of organisms.

We agree with the reviewer's comment that it is important to consider the effects of the added lysine residues when studying the functionality of proteins. We are aware that additional lysine residues may affect protein folding and function, and this might be considered as a limitation of the method. However, we do provide evidence that in two genes that we tested, chloramphenicol acetyltransferase (CAT) and phosphoribosylaminoimidazolesuccinocarboxamide synthase (ADE1, SAICAR), addition of 10 or 12 lysine residues has not affected protein function or stability based on our in vivo assays or western blot analysis (Fig. 5 and Sup. Fig. 13). These two proteins have different structures and belong to two different classes of proteins (based on SCOP database (<http://scop.mrc-lmb.cam.ac.uk/scop/>)). CAT is in the class of alpha and beta proteins with parallel beta sheets (a/b), with CoA-dependent acyltransferase fold, and ADE1(SAICAR) is in the class of alpha and beta proteins with antiparallel beta sheets (a+b), with SAICAR synthase like fold. Moreover, ADE1 protein is a monomer (PDB entry: 1A48) while CAT is a trimer (PDB; 3CLA) and needs to oligomerize to be functional (Leslie et al., 1988, PNAS). The fact that we observed a clear correlation between protein levels and activity as polyA sequences were added to these two structurally diverse proteins supports our method's utility. This, together with the rather predictable changes in expression of a rather diverse set of engineered and functional reporters as well as different fluorescent proteins and the human hemoglobin protein strongly supports our conclusion

that adding polyA tracks can produce hypomorphic alleles. We have now included additional description of genes that we used for our study in revised manuscript.

We have also discussed in the text that investigators interested in creating hypomorphic mutants could use the same number of lysine residues encoded by AAG codons as a control for the effects of poly-lysine tracks on protein stability and function. While we suggested to use polyA tracks (or control poly AAG sequence) as an N-terminal insertions, our experiments in Fig. 2 (fusion of MLP1-polyA track-YFP) and Sup. Fig. 9 (insertion of polyA track in the loop region (2nd exon) of human hemoglobin (HBD; also described in Arthur et al. 2015)) argue that polyA tracks (AAG controls) can be inserted either in the domain boundaries (MLP-YFP) or unstructured loops (HBD) of proteins without influencing protein function (YFP fluorescence) or protein stability (HBD, Arthur et al., 2015). As with any specific engineered allele, investigators will need to empirically test where they should insert control AAG and polyA track sequences as well as whether this method is suitable for the creation of hypomorphic mutants in the genes of interests. We have also implied in our discussion that for the 2% of genes that already have polyA tracks or many more genes with poly-lysine sequences, polyA tracks can be engineered through synonymous changes of AAG and AAA codons to create hypomorphic mutants.

Finally, the paper suggests (line 121-122) that the authors hypothesized that expression levels decrease with increasing polyA track length. However the two earlier papers (ref 16 & 17) already showed that this is the case in both eukaryotes and bacteria. Thus whilst this current paper does provide a more extensive study across a wider range of organisms, some of its main conclusions had already been reached in the two earlier publications. The paper/project could be strengthened by demonstrating a real application in gene functional analysis or synthetic biology, in addition to measuring expression levels of a few well know genes that are easy to quantify/analyse.

Author's response:

Our earlier publications provide a description of polyA tracks as an endogenous regulatory element in eukaryotic and prokaryotic genomes as well as the mechanism by which they can operate. In this method orientated manuscript, we demonstrate that insertion of polyA tracks is an effective tool to generate hypomorphic mutations to investigate gene function in multiple genes and model organisms. Our primary aim is to show that, in addition to incrementally decreasing protein expression, insertion of polyA tracks is independent of promoter strength; works at a similar level in all tested model organisms, and typically does not disrupt the function of the targeted gene or reporter protein. It is our thought that data on CAT and ADE1 gene regulation show actual studies on particular gene functions and/or synthetic biology application (at least in yeast this can be seen by a regulated dependence on adenine biosynthesis pathway). These are additional points to previously mentioned differences between CAT and ADE1 gene products. We have also included new data with additional sequence modifications of polyA tracks (insertion of other codons, or programming the length of polyA track with non-lysine codons) to show how polyA tracks can be further engineered for achieving fine tuning of hypomorphic

gene product amounts (Supplementary Fig. 2). We feel that these two examples show how the method described in this manuscript can be used for creation of hypomorphic alleles of genes of interest.

Reviewer #2 (Remarks to the Author):

Overall assessment

This novel, well-written manuscript describes the use of polynucleotide repeat tracts within gene coding regions for a range of cross-kingdom hosts to fine-tune gene expression through translational attenuation. I applaud the creative utilization of repeat tracts in this manner and am impressed to see its general applicability across a broad range of organisms.

I particularly like the concept of modifying existing genes that have poly-A tracts to fine-tune expression or to introduce these tracts with or without repeat degeneracy to introduce a method to control expression. These ideas are certainly a topic of interest to synthetic biologists and could be stressed more clearly in the introduction and discussion, especially for model organisms that currently lack reliable systems to tune gene expression.

Author's response:

We are thankful to the reviewer for his overall positive comments and interest in the presented method. Based on the suggestion, we have now expanded our discussion of applications of this method with emphasis on synthetic biology.

I do, however, have reservations about the reliability and reproducibility of the approach given the high rate of insertion-deletion mutations documented for long poly-nucleotide repeats (Natural variation: Brinkmann et al AJHG 1998, Ellegren Nat Rev Genet 2004; engineering / synthetic biology context: Egbert & Klavins, PNAS 2012). In addition to citing relevant work on repeat instability, I suggest the authors address the following items to rework the manuscript for publication.

Author's response:

We agree with the reviewer that the known hypermutability of long polynucleotide repeats should be cited and addressed with respect to insertion of polyA tracks. We have now included suggested citations and added a discussion of polyA track element mutation rate in the section of the manuscript that deals with the *D. melanogaster* model system as well as in the discussion (see further response below).

My main concern that is not addressed in the manuscript is the population heterogeneity that is certainly introduced by long repeat elements that are subject to high-rate insertion/deletion mutations. While the

discussion points out some challenges with existing expression optimization systems but fails to identify limitations of the authors' approach, specifically with regard to the hypermutable nature of long mono-nucleotide repeat tracts. Not addressing this concern could severely limit the impact of the authors' approach.

Many applications may require long-term expression of hypomorphic target genes over many generations, suggesting that mutations in a hypervariable poly-A tract could generate significant cell-cell genetic variability, limiting quantitative assessment of gene expression either spatially or temporally.

While the ultimate destination of this approach in most cases is single-copy chromosomal integration, many of the test conditions for the manuscript are transient transfections or plasmid transformations, each of which provide a population of target genes to individual cells. Multi-copy gene dosage may mask the effect of mutagenesis on repeat-modulated gene expression.

The expression attenuator design utilizes the poly-A tract either as an N-terminus tag or between two components of a fusion protein. In the case of the fusion protein, an insertion/deletion mutation to the repeat element may eliminate fluorescence but the N-terminus protein would remain functional. Thus, a fluorescent readout of attenuation does not necessarily indicate expression.

*I would like to see the authors address this concern by quantifying the mutation rate for one or more of the test systems. One simple test could be to perform flow cytometry of fluorescently labeled *E. coli* to identify what fraction of the population is non-fluorescent as a function of cell generation compared to controls, though this may require genomic integration to quantify if the plasmid copy number, and thus masking effect, is high.*

Author's response:

Based on the reviewer's comments, we have now assessed the cell population heterogeneity of our longest polyA track insertion (36As) in the *D. melanogaster* genome. We felt that this system and model organism would be the best to get new data for mutation rates of polyA tracks. Genomic DNA was isolated from a whole adult fly after more than 30 generations of crosses between homozygotes with insertion of 36As polyA track in the mCherry reporter gene. The region of interest was amplified under the same conditions for both the isolated genomic DNA and the original DNA construct that was used for generation of the transgenic insect. PCR products were ligated to Illumina adapters and subjected to sequencing. Illumina sequencing of the isolated polyA track (36As) regions from the fruit fly genome and original DNA vector revealed less than 10% difference between the two data sets. Mutations in genomic data include complete loss of the polyA track (approx. 8%, this can be assigned to the recombination events and similarity to fruit flies chromosome X and 3, supplementary Figure 9). Since Illumina sequencing is not reliable for homopolymers we have additionally performed Sanger sequencing on the same set of sequences (approx. 100 colonies each) revealed that insertions or polyA track shortening indeed occurs at low frequency. These results have been added as Supplementary Fig. 9

and a short description has been added in the *D. melanogaster* section as well as in the discussion and methods of the manuscript. These results are in the range of already described 7% hypermutability of the polyA(/T) tracks in the human germline mutations of BAT-40 microsatellite (40As) located in the second intron of the 3-beta-hydroxysteroid dehydrogenase gene (Bacon et al., NAR, 2001). We have to note that our data show general mutation rates for the whole organism while in the case of mentioned study (Bacon et al., NAR, 2001) the mutation rate is dependent on the cell type and it is typically lower for somatic cells. We also point out to the study by Chung et al., (2008, PLoS One). The authors find that the mutation rate in polyA region (10As in this case) is in the range of 10^{-4} per cell per generation. The simple calculation argues that approx. 1% of cells will be affected by a mutation in 100 generations. This is also consistent with the data from Egbert & Klavins (PNAS, 2012) as well as with mutation rates generally reported for short tandem repeats (STRs) from multiple manuscripts (approximately 10^{-6} – 10^{-2} mutation rate). In the end, one should keep in mind that polyA tracks tend to operate on the shorter side of length distribution of STRs and, as such, data from STRs could be treated with caution. We also feel that mutability of the certain regions might be different and we would not like to make strong statements about the possible mutation rates of homopolymers in different systems. This can be explored empirically by investigators.

Secondary questions

Line 348 (discussion): how is this method necessarily rapid? It seems that simple, rather than rapid, better describes the approach. I agree that simple, predictable mapping of sequence to expression may allow rapid pinpointing of optimal expression levels. The manuscript could be more clear in assessing the uniqueness of the approach.

Author's response:

We have expanded the discussion of the uniqueness of this method. We feel that this method is rapid because it does not rely on identification of hypomorphic alleles through random mutation and screen approaches.

Have the authors attempted to use of other repeat elements to attain greater control over gene expression? Poly-T or poly-AT tracts, for instance? Some treatment of the feasibility of extensions with other repeats in the discussion could help frame the outlook for finer control or additional applications.

Author's response: We have focused our method on polyA tracks due to our previous studies on the mechanism of polyA track induced ribosome stalling and frameshifting (Arthur et al., 2015 and Koutmou et al., 2015). The use of polyT or polyAT tracks would be severely limited due to several factors (DNA and RNA polymerase slippage (polyT), tRNA abundance (rare Ile ATA codon for several organisms) as well as strong selection against consecutive hydrophobic residues in the protein structures (polyT results in consecutive Phe residues or polyAT in consecutive Ile-Tyr pairs). We have now added Supplementary Fig. 2 with additional sequence modifications of polyA tracks (insertion of other nucleotides/codons, or

programming the length of polyA track with non-lysine codons) to show how polyA tracks can be further engineered for achieving fine tuning of hypomorphic gene product amounts.

For figure 2d-f, the (AAG)₁₂ values for protein and mRNA levels are not consistent. The protein band should be 60% of WT based on qPCR data, yet it hardly appears on the western blot. Panel F appears to be the outlier, while D & E appear consistent. The inconsistency should be addressed in the text or the experiments redone for clarify.

In figure 3, the results for (AAG)₁₂ in panels c and d again appear to be inconsistent. The qPCR results appear similar to wild-type but the protein levels are significantly lower.

The text addresses these concerns in part in lines 219-222, but this seems buried or overlooked. The authors could consider reworking the text or expanding the treatment of the apparent inconsistencies to help clarify the results.

Author's response:

We have expanded the discussion of polybasic peptide induced stalling (induced by AAG codons) and clarified the inconsistencies in the text where needed. The results from different cells indicate our previous observation that the polybasic peptide stalling (induced with 12 lysine AAG codons) operates through a different mechanism than polyA track induced ribosome stalling and frameshifting. One of the consequences of these different pathways is the discrepancy between mRNA and protein levels in the case of polybasic stalling. However, it is our general observation that effects of polyA tracks show more consistent relationship between mRNA levels and protein accumulation levels across multiple organisms and cells compared to polybasic stalling.

Line 261 - seems it should reference Figure 4, not Figure 3.

Line 271 concentration listed for Dox is not consistent with Figure 5 or Figure 5 caption

Line 275 - Relative protein for Figure 5b appears 12-17% WT, inconsistent with text

Figure 5 caption, panel b - it could be more clear here whether the percent WT measure in (b)/(c) is calculated for each Dox level or a single value (such as at max Dox) to normalize all samples. I assume the former to show a consistent fraction of expression at different induction levels, but some more textual clarity would help.

Lines 563-569 - do "dox concentration" for (b) and "final dox concentration" for (c) mean the same thing here? If so, unify the text descriptions.

Supplement line 93 - "was monitored"?

Caption for Supp fig 7 - standard deviation at end of caption appears truncated to "standard"

Author's response:

We are thankful for the reviewer for pointing out these mistakes. Corrections are made in the revised manuscript.

Reviewers' Comments:

Reviewer #1 (Remarks to the Author)

The authors agree that the additional Lys residues can effect protein function and this is a potential limitation of their approach. However, they did not discuss this further in the revised paper. I suggest that in the discussion section, after the point where they describe the limitations of the competing methods (mutagenesis of SD sequences & orthogonal translations systems) that they also describe the limitations of their approach more clearly.

The revised paper included additional data for fine tuning expression levels (Supplementary Fig. 2) which was useful, but there were no further demonstrations of applications in synthetic biology etc. Nevertheless, at end of the paper more suggestions for possible applications have been described. I would therefore suggest publication, subject to minor revision, so that the community can begin to explore more fully the utility of this approach.

Reviewer #2 (Remarks to the Author)

The authors' response is very thorough and addresses all of our comments. We have no further concerns that would prevent me from recommending this manuscript for publication in Nature Communications with no further revisions.

Reviewer #1 (Remarks to the Author):

The authors agree that the additional Lys residues can affect protein function and this is a potential limitation of their approach. However, they did not discuss this further in the revised paper. I suggest that in the discussion section, after the point where they describe the limitations of the competing methods (mutagenesis of SD sequences & orthogonal translations systems) that they also describe the limitations of their approach more clearly.

The revised paper included additional data for fine tuning expression levels (Supplementary Fig. 2) which was useful, but there were no further demonstrations of applications in synthetic biology etc. Nevertheless, at end of the paper more suggestions for possible applications have been described. I would therefore suggest publication, subject to minor revision, so that the community can begin to explore more fully the utility of this approach.

Author's response:

We are thankful to the reviewer for his positive attitude and valuable comments that have further improved our manuscript. We have now included in the Discussion section of the manuscript a part of our previous response to the reviewer concerning the insertion of the lysine residues and possible limitation.

The paragraph states:

“The addition of a polyA track to the target gene will result in additional lysine residues in the protein product. Like any protein tag, it is important to consider the effects of the additional residues when studying the functionality of the protein. We have shown that the function and stability of two structurally diverse proteins, CAT and Ade1, are not affected by up to 12 additional lysine residues. To control for possible effects of the poly-lysine tracks, investigators can create an allele with the same number of lysine residues encoded by AAG codons. The AAG codons will have minimal effect on expression levels while encoding a synonymous protein. Furthermore, the flexibility in polyA track placement within the coding sequence allows investigators to choose the most suitable insertion site for the protein of interest. “

Reviewer #2 (Remarks to the Author):

The authors' response is very thorough and addresses all of our comments. We have no further concerns that would prevent me from recommending this manuscript for publication in Nature Communications with no further revisions.

Author's response:

We appreciate the reviewer's recommendation that our manuscript be published in Nature communication with no further revisions. We also appreciate the reviewers previous comments that we feel have helped to greatly improve the manuscript.